# Robust Multi-Agent Reinforcement Learning with State Uncertainty

**Sihong He**                                                                 *sihong.he@uconn.edu*
*Department of Computer Science and Engineering*
*University of Connecticut*

**Songyang Han**                                                          *songyang.han@uconn.edu*
*Department of Computer Science and Engineering*
*University of Connecticut*

**Sanbao Su**                                                                 *sanbao.su@uconn.edu*
*Department of Computer Science and Engineering*
*University of Connecticut*

**Shuo Han**                                                                      *hanshuo@uic.edu*
*Department of Electrical and Computer Engineering*
*University of Illinois, Chicago*

**Shaofeng Zou**                                                               *szou3@buffalo.edu*
*Department of Electrical Engineering*
*University at Buffalo, The State University of New York*

**Fei Miao**                                                                     *fei.miao@uconn.edu*
*Department of Computer Science and Engineering*
*University of Connecticut*

**Reviewed on OpenReview:** *https://openreview.net/forum?id=CqTkapZ6H9*

## Abstract

In real-world multi-agent reinforcement learning (MARL) applications, agents may not have perfect state information (e.g., due to inaccurate measurement or malicious attacks), which challenges the robustness of agents' policies. Though robustness is getting important in MARL deployment, little prior work has studied state uncertainties in MARL, neither in problem formulation nor algorithm design. Motivated by this robustness issue and the lack of corresponding studies, we study the problem of MARL with state uncertainty in this work. We provide the first attempt to the theoretical and empirical analysis of this challenging problem. We first model the problem as a Markov Game with state perturbation adversaries (MG-SPA) by introducing a set of state perturbation adversaries into a Markov Game. We then introduce robust equilibrium (RE) as the solution concept of an MG-SPA. We conduct a fundamental analysis regarding MG-SPA such as giving conditions under which such a robust equilibrium exists. Then we propose a robust multi-agent Q-learning (RMAQ) algorithm to find such an equilibrium, with convergence guarantees. To handle high-dimensional state-action space, we design a robust multi-agent actor-critic (RMAAC) algorithm based on an analytical expression of the policy gradient derived in the paper. Our experiments show that the proposed RMAQ algorithm converges to the optimal value function; our RMAAC algorithm outperforms several MARL and robust MARL methods in multiple multi-agent environments when state uncertainty is present. The source code is public on `https://github.com/sihongho/robust_marl_with_state_uncertainty`.

# 1 Introduction

Reinforcement Learning (RL) recently has achieved remarkable success in many decision-making problems, such as robotics, autonomous driving, traffic control, and game playing (Espeholt et al., 2018; Silver et al., 2017; Mnih et al., 2015; He et al., 2022). However, in real-world applications, the agent may face *state uncertainty* in which accurate information about the state is unavailable. This uncertainty may be caused by unavoidable sensor measurement errors, noise, missing information, communication issues, and/or malicious attacks. A policy not robust to state uncertainty can result in unsafe behaviors and even catastrophic outcomes. For instance, consider the path planning problem shown in Figure 1, where the agent (green ball) observes the position of an obstacle (red ball) through sensors and plans a safe (no collision) and shortest path to the goal (black cross). In Figure 1-(a), the agent can observe the true state $s$ (red ball) and choose an optimal and collision-free curve $a^*$ (in red) tangent to the obstacle. In com-

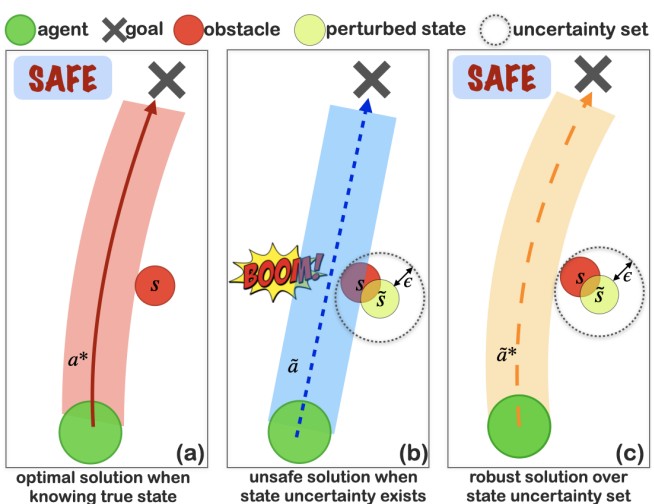

Figure 1: Motivation of considering state uncertainty in single-agent reinforcement learning.

parison, when the agent can only observe the perturbed state $\tilde{s}$ (yellow ball) caused by inaccurate sensing or state perturbation adversaries (Figure 1-(b)), it will choose a straight line $\tilde{a}$ (in blue) as the shortest and collision-free path tangent to $\tilde{s}$. However, by following $\tilde{a}$, the agent actually crashes into the obstacle. To avoid collision in the worst case, one can construct a state uncertainty set that contains the true state based on the observed state. Then the robustly optimal path under state uncertainty becomes the yellow curve $\tilde{a}^*$ tangent to the uncertainty set, as shown in Figure 1-(c).

In single-agent RL, imperfect information about the state has been studied in the literature of partially observable Markov decision process (POMDP) (Kaelbling et al., 1998). However, as pointed out in recent literature (Huang et al., 2017; Kos & Song, 2017; Yu et al., 2021b; Zhang et al., 2020a), the conditional observation probabilities in POMDP cannot capture the *worst-case* (or adversarial) scenario, and the learned policy without considering state uncertainties may fail to achieve the agent's goal. Dealing with state uncertainty becomes even more challenging for Multi-Agent Reinforcement Learning (MARL), where each agent aims to maximize its own total return during the interaction with other agents and the environment (Yang & Wang, 2020b). Even if one agent receives misleading state information, its action affects both its own return and the other agents' returns (Zhang et al., 2020b) and may result in catastrophic failure. The existing literature of decentralized partially observable Markov decision process (Dec-POMDP) (Oliehoek et al., 2016) does not provide theoretical analysis or algorithmic tools for MARL under worst-case state uncertainties either.

To better illustrate the effect of state uncertainty in MARL, the path planning problem in Figure 1 is modified such that two agents are trying to reach their individual goals without collision (a penalty or negative reward applied). When the blue agent knows the true position $s_0^g$ (the subscript denotes time, which starts from 0) of the green agent, it will get around the green agent to quickly reach its goal without collision. However, in Figure 2-(a), when the blue agent can only observe the perturbed position $\tilde{s}_0^g$ (yellow circle) of the green agent, it would choose a straight line that it thought safe (Figure 2-(a1)), which eventually leads to a crash (Figure 2-(a2)). In Figure 2-(b), the blue agent adopts a robust trajectory by considering a state uncertainty set based on its observation. As shown in Figure 2-(b1), there is no overlap between $(s_0^b, \tilde{s}_0^g)$ or $(s_T^b, \tilde{s}_T^g)$. Since the uncertainty sets centered at $\tilde{s}_0^g$ and $\tilde{s}_T^g$ (the dotted circles) include the true state of the green agent, this robust trajectory also ensures no collision between $(s_0^b, s_0^g)$ or $(s_T^b, s_T^g)$. The blue agent considers the interactions with the green agent to ensure no collisions at any time. Therefore, it is necessary to consider state uncertainty in a multi-agent setting where the dynamics of other agents should be considered.

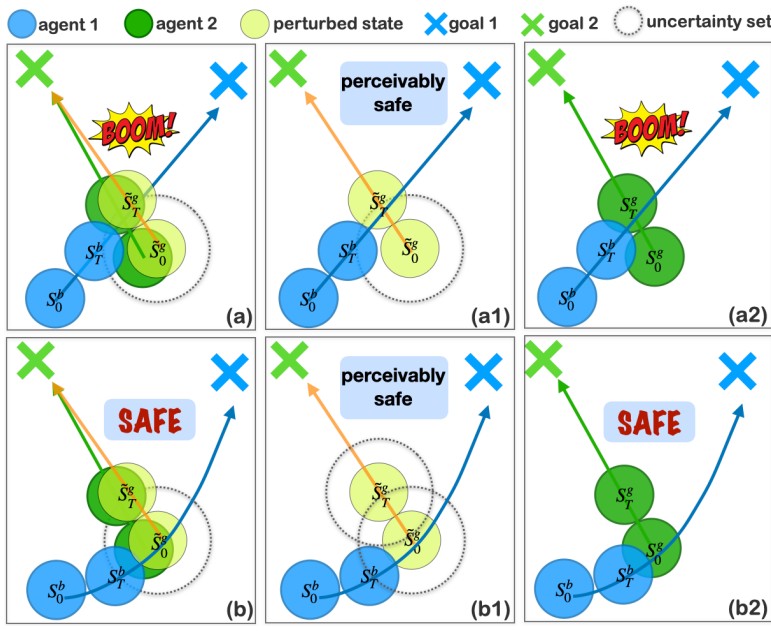

Figure 2: Motivation of considering state uncertainty in MARL.

In this work, we develop a robust MARL framework that accounts for state uncertainty. Specifically, we model the problem of MARL with state uncertainty as a Markov game with state perturbation adversaries (MG-SPA), in which each agent is associated with a state perturbation adversary. One state perturbation adversary always plays against its corresponding agent by preventing the agent from knowing the true state accurately. We analyze the MARL problem with adversarial or worst-case state perturbations. Compared to single-agent RL, MARL is more challenging due to the interactions among agents and the necessity of studying equilibrium policies (Nash, 1951; McKelvey & McLennan, 1996; Slantchev, 2008; Daskalakis et al., 2009; Etessami & Yannakakis, 2010). The contributions of this work are summarized as follows.

**Contributions:** To the best of our knowledge, this work is the first attempt to systematically characterize state uncertainties in MARL and provide both theoretical and empirical analysis. First, we formulate the MARL problem with state uncertainty as a Markov game with state perturbation adversaries (MG-SPA). We define the solution concept of the game as a robust equilibrium (RE), where all players including the agents and the adversaries use policies from which no one has an incentive to deviate. In an MG-SPA, each agent not only aims to maximize its return when considering other agents' actions but also needs to act against all state perturbation adversaries. Therefore, a robust equilibrium policy of one agent is robust to state uncertainties. Second, we study its fundamental properties and prove the existence of a robust equilibrium under certain conditions. We develop a robust multi-agent Q-learning (RMAQ) algorithm with a convergence guarantee and a robust multi-agent actor-critic (RMAAC) algorithm for handling high-dimensional state-action space. Finally, we conduct experiments in a two-player game to validate the convergence of the proposed Q-learning method RMAQ. We test our RMAAC algorithm in several benchmark multi-agent environments. We show that our RMAQ and RMAAC algorithms can learn robust policies that outperform baselines under state perturbations in multi-agent environments.

**Organization:** The rest of the paper is organized as follows. The related work is presented in Section 2. In Section 3, we introduce some preliminary concepts in RL and MARL. The proposed methodology and corresponding analysis are in Section 4. The proposed algorithms are in Section 5 and experiments results are in Section 6. We discuss some future work in Section 7. In Section 8 we conclude.

## 2 Related work

**Robust Reinforcement Learning:** Recent robust reinforcement learning studied different types of uncertainties, such as action uncertainties (Tessler et al., 2019) and transition kernel uncertainties (Sinha et al., 2020; Yu et al., 2021b; Hu et al., 2020; Wang & Zou, 2021; Lim & Autef, 2019; Nisioti et al., 2021; He et al., 2022). Some recent attempts at adversarial state perturbations for single-agent validated the importance of considering state uncertainty and improving the robustness of the learned policy in Deep RL (Huang et al., 2017; Lin et al., 2017; Zhang et al., 2020a; 2021; Everett et al., 2021). The works of Zhang et al. (2020a; 2021) formulate the state perturbation in single-agent RL as a modified Markov decision process, then study the robustness of single-agent RL policies. The works of Huang et al. (2017) and Lin et al. (2017) show that adversarial state perturbation undermines the performance of neural network policies

in single-agent reinforcement learning and proposes different single-agent attack strategies. In this work, we consider the more challenging problem of adversarial state perturbation for MARL, when the environment of an individual agent is non-stationary with other agents' changing policies during the training process.

**Robust Multi-Agent Reinforcement Learning:** There is very limited literature on the solution concept or theoretical analysis when considering adversarial state perturbations in MARL. Other types of uncertainties have been investigated in the literature, such as uncertainties about training partner's type (Shen & How, 2021), the other agents' policies (Li et al., 2019; Sun et al., 2021; van der Heiden et al., 2020), and reward uncertainties (Zhang et al., 2020b). However, the policy considered in these papers relies on the true state information. Hence, the robust MARL considered in this work is fundamentally different since the agents do not know the true state information. Dec-POMDP enables a team of agents to optimize policies with the partial observable states (Oliehoek et al., 2016; Chen et al., 2022). The work of Lin et al. (2020) studies state perturbation in identical-interest MARL, and proposes an attack method to attack the state of one single agent in order to decrease the team reward. In contrast, we consider the worst-case scenario that the state of every agent can be perturbed by an adversary and focus on the theoretical analysis of robust MARL including the existence of optimal value function and robust equilibrium (RE). Our work provides formal definitions of the state uncertainty challenge in MARL, and derives both theoretical analysis and practical algorithms.

**Game Theory and MARL:** MARL shares theoretical foundations with the game theory research field and a literature review has been provided to understand MARL from a game theoretical perspective (Yang & Wang, 2020a). A Markov game, sometimes called a stochastic game models the interaction between multiple agents (Owen, 2013; Littman, 1994). Algorithms to compute the Nash equilibrium (NE) in Dec-POMDP (Oliehoek et al., 2016), POSG (partially observable stochastic game) and analysis assuming that NE exists (Chades et al., 2002; Hansen et al., 2004; Nair et al., 2002) have been developed in the literature without proving the conditions for the existence of NE. The main theoretical contributions of this work include proving conditions under which the proposed MG-SPA has robust equilibrium solutions, and convergence analysis of our proposed robust multi-agent Q-learning algorithm. This is the first attempt to analyze the fundamental properties of MARL under adversarial state uncertainties.

## 3 Preliminary

**Q-learning** is a model-free single-agent reinforcement learning algorithm (Sutton et al., 1998). The core of this method is a Bellman equation that $q^*(s, a) = r(s, a) + \gamma \sum_{s'} \max_{a' \in A} q^*(s', a')$. The Bellman equation encourages a simple value iteration update which uses the weighted average of old Q-value and the new one. Q-learning learns the optimal action-value function $q_*(s, a)$ by value iteration: $q_{new}(s, a) = (1 - \alpha)q_{old}(s, a) + \alpha [r(s, a) + \gamma \sum_{s'} p(s'|s, a) \max_{a' \in A} q_{old}(s', a')]$, and the optimal action $a_*(s) = \arg\max_{a \in A} q_*(s, a)$. Deep Q-Networks (DQN) use a neural network with parameter $\theta$ to approximate Q-value (Mnih et al., 2015). This allows the algorithm to handle larger state spaces. DQN minimizes the loss function defined in (1), where $q'$ is a target network that copies the parameter $\theta$ occasionally to make training more stable. $\mathcal{D}$ is an experience replay buffer. DQN uses experience replay, which involves storing and randomly sampling previous experiences to train the neural network, to improve the stability and efficiency of the learning process. The target network helps to prevent the algorithm from oscillating or diverging during training.

$$\mathcal{L}(\theta) = \mathbb{E}_{\tau \sim \mathcal{D}} \left[ y - q(s, a|\theta) \right]^2, \quad y = r(s, a) + \gamma \max_{a' \in A} q'(s', a'). \tag{1}$$

**Actor-Critic (AC)** is a single-agent reinforcement learning algorithm with two parts: an actor decides which action should be taken and a critic evaluates how well the actor performs (Sutton et al., 1998). The actor is parameterized by $\pi_\theta(\cdot|s)$ and iteratively updates the parameter $\theta$ to maximize the objective function $J(\theta) = \mathbb{E}_{\tau \sim p, a \sim \pi_\theta}[\sum_{t=1}^{\infty} \gamma^{t-1} r_t(s_t, a_t)]$, where $\tau$ denotes a trajectory and $p$ is the state transition probability distribution. The critic is parameterized by $q_\phi(s, a)$ and evaluates actions chosen by the actor by computing the Q-value i.e. action-value function. The critic can update itself by using (1). The actor updates its parameter by using the gradient: $\nabla_\theta J(\theta) = \mathbb{E}_{s \sim p, a \sim \pi_\theta} [q^\pi(s, a) \nabla_\theta \log \pi_\theta(a|s)]$.

**Markov game (MG)** is used to model the interaction between multiple agents (Littman, 1994). A Markov game, sometimes is called a stochastic game (Owen, 2013) defined as a tuple $G := (\mathcal{N}, S, \{A^i\}_{i \in \mathcal{N}}, \{r^i\}_{i \in \mathcal{N}}, p, \gamma)$, where $S$ is the state space, $\mathcal{N}$ is a set of $N$ agents, $A^i$ is the action space of agent $i$, respectively (Littman, 1994; Owen, 2013). $\gamma \in [0, 1)$ is the discounting factor. We define $A = A^1 \times \cdots \times A^N$ as the joint action space. The state transition $p : S \times A \to \Delta(S)$ is controlled by the current state and joint action, where $\Delta(S)$ represents the set of all probability distributions over the joint state space $S$. Each agent has a reward function, $r^i : S \times A \to \mathbb{R}$. At time $t$, agent $i$ chooses its action $a_t^i$ according to a policy $\pi^i : S \to \Delta(A^i)$. For each agent $i$, it attempts to maximize its expected sum of discounted rewards, i.e. its objective function $J^i(s, \pi) = \mathbb{E}\left[\sum_{t=1}^{\infty} \gamma^{t-1} r_t^i(s_t, a_t) | s_1 = s, a_t \sim \pi(\cdot | s_t)\right]$.

## 4 Methodology

In this section, to solve the robust multi-agent reinforcement learning problem with state uncertainty, we first introduce the framework of the Markov game with state perturbation adversaries. We then provide characterization results for the proposed framework: Markov and history-dependent policies, the definition of a solution concept called robust equilibrium based on value functions, derivation of Bellman equations, as well as certain conditions for the existence of a robust equilibrium and the optimal value function.

### 4.1 Markov Game with State Perturbation Adversaries

We use a tuple $\tilde{G} := (\mathcal{N}, \mathcal{M}, S, \{A^i\}_{i \in \mathcal{N}}, \{B^{\tilde{i}}\}_{\tilde{i} \in \mathcal{M}}, \{r^i\}_{i \in \mathcal{N}}, p, f, \gamma)$ to denote a Markov game with state perturbation adversaries (MG-SPA). In an MG-SPA, we introduce an additional set of adversaries $\mathcal{M} = \{\tilde{1}, \cdots, \tilde{N}\}$ to a Markov game (MG) with an agent set $\mathcal{N}$. Each agent $i$ is associated with an adversary $\tilde{i}$ and can observe the true state $s \in S$ if without adversarial perturbation. Each adversary $\tilde{i}$ is associated with an action $b^{\tilde{i}} \in B^{\tilde{i}}$ and the same state $s \in S$ that agent $i$ has. We define the adversaries' joint action as $b = (b^{\tilde{1}}, ..., b^{\tilde{N}}) \in B$, $B = B^{\tilde{1}} \times \cdots \times B^{\tilde{N}}$. At time $t$, adversary $\tilde{i}$ can manipulate the corresponding agent $i$'s state information. Once adversary $\tilde{i}$ gets state $s_t$, it chooses an action $b_t^{\tilde{i}}$ according to a policy $\rho^{\tilde{i}} : S \to \Delta(B^{\tilde{i}})$. According to a perturbation function $f$, adversary $\tilde{i}$ perturbs state $s_t$ to $\tilde{s}_t^i = f(s_t, b_t^{\tilde{i}}) \in S$. We use $\tilde{s}_t = (\tilde{s}_t^1, \cdots, \tilde{s}_t^N)$ to denote a joint perturbed state and use the notation $f(s_t, b_t) = \tilde{s}_t$. We denote the adversaries' joint policy as $\rho(b|s) = \prod_{\tilde{i} \in \mathcal{M}} \rho^{\tilde{i}}(b^{\tilde{i}}|s)$. The definitions of agent action and agents' joint action are the same as their definitions in an MG. Agent $i$ chooses its action $a_t^i$ with $\tilde{s}_t^i$ according to a policy $\pi^i(a_t^i|\tilde{s}_t^i)$, $\pi^i : S \to \Delta(A^i)$. We denote the agents' joint policy as $\pi(a|\tilde{s}) = \prod_{i \in \mathcal{N}} \pi(a^i|\tilde{s}^i)$. Agents execute the agents' joint action $a_t$, then at time $t+1$, the joint state $s_t$ turns to the next state $s_{t+1}$ according to a transition probability function $p : S \times A \times B \to \Delta(S)$. Each agent $i$ gets a reward according to a state-wise reward function $r_t^i : S \times A \times B \to \mathbb{R}$. Each adversary $\tilde{i}$ gets an opposite reward $-r_t^i$. In an MG, the transition probability function and reward function are considered as the model of the game. In an MG-SPA, the perturbation function $f$ is also considered as a part of the model, i.e., the model of an MG-SPA consists of $f, p$ and $\{r^i\}_{i \in \mathcal{N}}$.

---

**Definition 4.1** (Value Functions)**.**

$v^{\pi, \rho} = (v^{\pi, \rho, 1}, \cdots, v^{\pi, \rho, N}), q^{\pi, \rho} = (q^{\pi, \rho, 1}, \cdots, q^{\pi, \rho, N})$ *are defined as the state-value function or value function for short, and the action-value function, respectively. The ith element $v^{\pi, \rho, i}$ and $q^{\pi, \rho, i}$ are defined as following:*

$$q^{\pi, \rho, i}(s, a, b) = \mathbb{E}\left[\sum_{t=1}^{\infty} \gamma^{t-1} r_t^i | s_1 = s, a_1 = a, b_1 = b, a_t \sim \pi(\cdot | \tilde{s}_t), b_t \sim \rho(\cdot | s_t), \tilde{s}_t = f(s_t, b_t)\right], \quad (2)$$

$$v^{\pi, \rho, i}(s) = \mathbb{E}\left[\sum_{t=1}^{\infty} \gamma^{t-1} r_t^i | s_1 = s, a_t \sim \pi(\cdot | \tilde{s}_t), b_t \sim \rho(\cdot | s_t), \tilde{s}_t = f(s_t, b_t)\right]. \quad (3)$$

---

To incorporate realistic settings into our analysis, we restrict the power of each adversary, which is a common assumption for state perturbation adversaries in the RL literature (Zhang et al., 2020a; 2021; Everett et al., 2021). We define perturbation constraints $\tilde{s}^i \in \mathcal{B}_{dist}(\epsilon, s) \subset S$ to restrict the adversary $\tilde{i}$ to perturb a

state only to a predefined set of states. $\mathcal{B}_{dist}(\epsilon, s)$ is a $\epsilon$-radius ball measured in metric $dist(\cdot, \cdot)$, which is often chosen to be the $l$-norm distance: $dist(s, \tilde{s}^i) = \|s - \tilde{s}^i\|_l$. We omit the subscript $dist$ in the following context. For each agent $i$, it attempts to maximize its expected sum of discounted rewards, i.e. its objective function $J^i(s, \pi, \rho) = \mathbb{E}\left[\sum_{t=1}^{\infty} \gamma^{t-1} r_t^i(s_t, a_t) | s_1 = s, a_t \sim \pi(\cdot|\tilde{s}_t), \tilde{s}_t = f(s_t, b_t), b_t \sim \rho(\cdot|s_t)\right]$. Each adversary $\tilde{i}$ aims to minimize the objective function of agent $i$ and is considered as receiving an opposite reward of agent $i$, which also leads to a value function $-J^i(s, \pi, \rho)$ for adversary $\tilde{i}$. We further define the value functions in an MG-SPA as in Definition 4.1. Then we propose robust equilibrium (RE), a NE-structured solution as our solution concept for the proposed MG-SPA framework. We formally define RE in Definition 4.2.

---

**Definition 4.2** (Robust Equilibrium).

*Given a Markov game with state perturbation adversaries $\tilde{G}$, a joint policy $d_* = (\pi_*, \rho_*)$ where $\pi_* = (\pi_*^1, \cdots, \pi_*^N)$ and $\rho_* = (\rho_*^{\tilde{1}}, \cdots, \rho_*^{\tilde{N}})$ is said to be in robust equilibrium, or a robust equilibrium, if and only if, for any $i \in \mathcal{N}$, $\tilde{i} \in \mathcal{M}$, $s \in S$,*

$$v^{(\pi_*^{-i}, \pi_*^i, \rho_*^{-\tilde{i}}, \rho^{\tilde{i}}), i}(s) \geq v^{(\pi_*^{-i}, \pi_*^i, \rho_*^{-\tilde{i}}, \rho_*^{\tilde{i}}), i}(s) \geq v^{(\pi_*^{-i}, \pi^i, \rho_*^{-\tilde{i}}, \rho_*^{\tilde{i}}), i}(s), \tag{4}$$

*where $-i/-\tilde{i}$ represents the indices of all agents/adversaries except agent $i$/ adversary $\tilde{i}$.*

---

As the maximin solution is also a popular solution concept in robust RL problems (Zhang et al., 2020a), here, we discuss why we choose a NE-structured solution other than a maximin solution. Firstly, we aim to propose a framework that can describe and model the interactions among agents when each agent has its own interest or reward function under state perturbations, and maximin solution may not be a general solution concept for robust MARL problems. A maximin solution is natural to use when considering robustness in single-agent RL problems and identical-interest MARL problems, but it fails to handle those MARL problems where each agent has its own interest or reward function. Secondly, the Nash equilibrium is also a commonly used robust solution concept in both single-agent RL and MARL problems (Tessler et al., 2019; Zhang et al., 2020b). Many papers have used NE as their solution concept to investigate robustness in RL problems, and to the best of our knowledge, the NE-structured solution is the only one used in robust non-identical-interest MARL problems (Zhang et al., 2020b). Lastly, for finite two-agent zero-sum games, it is known that NE, minimax, and maximin solution concepts all give the same answer (Yin et al., 2010; Owen, 2013).

After defining RE, we seek to characterize the optimal value $v_*(s) = (v_*^1(s), \cdots, v_*^N(s))$ defined by $v_*^i(s) = \max_{\pi^i} \min_{\rho^{\tilde{i}}} v^{(\pi_*^{-i}, \pi^i, \rho_*^{-\tilde{i}}, \rho^{\tilde{i}}), i}(s)$. For notation convenience, we use $v^i(s)$ to denote $v^{(\pi_*^{-i}, \pi^i, \rho_*^{-\tilde{i}}, \rho^{\tilde{i}}), i}(s)$. The Bellman equations of an MG-SPA are in the forms of (5) and (6). The Bellman equation is a recursion for expected rewards, which helps us identify or find an RE.

$$q_*^i(s, a, b) = r^i(s, a, b) + \gamma \sum_{s' \in S} p(s'|s, a, b) \max_{\pi^i} \min_{\rho^{\tilde{i}}} \mathbb{E}\left[q_*^i(s', a', b') | a' \sim \pi(\cdot|\tilde{s}), b' \sim \rho(\cdot|s)\right], \tag{5}$$

$$v_*^i(s) = \max_{\pi^i} \min_{\rho^{\tilde{i}}} \mathbb{E}\left[\sum_{s' \in S} p(s'|s, a, b)[r^i(s, a, b) + \gamma v_*^i(s')] | a \sim \pi(\cdot|\tilde{s}), b \sim \rho(\cdot|s)\right], \tag{6}$$

for all $i \in \mathcal{N}$, $\tilde{i} \in \mathcal{M}$, where $\pi = (\pi^i, \pi_*^{-i})$, $\rho = (\rho^{\tilde{i}}, \rho_*^{-\tilde{i}})$, and $(\pi_*^i, \pi_*^{-i}, \rho_*^{\tilde{i}}, \rho_*^{-\tilde{i}})$ is a robust equilibrium for $\tilde{G}$. We prove them in the following subsection. The policies in(5) and (6) are defined to be Markov policies which only input the current state. The robust equilibrium is also based on Markov policies. History-dependent policies for MARL under state perturbations may improve agents' ability to adapt to adversarial state perturbations and random sensor noise, by allowing agents to take into account past observations when making decisions. Therefore, we further discuss how the current MG-SPA frame and solution concept adapt to history-dependent policies in subsection 4.3.

## 4.2 Theoretical Analysis of MG-SPA

In this subsection, we first introduce the vector notations we used in the theoretical analysis and define a minimax operator in Definition 4.3. We then introduce Assumption 4.4 which is considered in our theoretical

analysis. Later, we prove two propositions about the minimax operator and Theorem 4.7 which shows a series of fundamental characteristics of an MG-SPA under Assumption 4.4, e.g., the derivation of Bellman equations, the existence of optimal value functions and robust equilibrium.

**Vector Notations:** To make the analysis easy to read, we follow and extend the vector notations in Puterman (2014). Let $V$ denote the set of bounded real valued functions on $S$ with component-wise partial order and norm $\|v^i\| := \sup_{s \in S} |v^i(s)|$. Let $V_M$ denote the subspace of $V$ of Borel measurable functions. For discrete state space, all real-valued functions are measurable so that $V = V_M$. But when $S$ is a continuum, $V_M$ is a proper subset of $V$. Let $v = (v^1, \cdots, v^N) \in \mathbb{V}$ be the set of bounded real valued functions on $S \times \cdots \times S$, i.e. the across product of $N$ state set and norm $\|v\| := \sup_j \|v^j\|$. For discrete $S$, let $|S|$ denote the number of elements in $S$. Let $r^i$ denote a $|S|$-vector, with $s$th component $r^i(s)$ which is the expected reward for agent $i$ under state $s$. And $P$ the $|S| \times |S|$ matrix with $(s, s')$th entry given by $p(s'|s)$. We refer to $r^i_d$ as the reward vector of agent $i$, and $P_d$ as the probability transition matrix corresponding to a joint policy $d = (\pi, \rho)$. $r^i_d + \gamma P_d v^i$ is the expected total one-period discounted reward of agent $i$, obtained using the joint policy $d = (\pi, \rho)$. Let $z$ as a list of joint policy $\{d_1, d_2, \cdots\}$ and $P^0_z = I$, we denote the expected total discounted reward of agent $i$ using $z$ as $v^i_z = \sum_{t=1}^{\infty} \gamma^{t-1} P^{t-1}_z r^i_{d_t} = r^i_{d_1} + \gamma P_{d_1} r^i_{d_2} + \cdots + \gamma^{n-1} P_{d_1} \cdots P_{d_{n-1}} r^i_{t_n} + \cdots$. Now, we define the following minimax operator which is used in the rest of the paper.

---

**Definition 4.3** (Minimax Operator)**.**

*For $v^i \in V, s \in S$, we define the nonlinear operator $L^i$ on $v^i(s)$ by $L^i v^i(s) := \max_{\pi^i} \min_{\rho^{\tilde{i}}} [r^i_d + \gamma P_d v^i](s)$, where $d := (\pi^{-i}_*, \pi^i, \rho^{-\tilde{i}}_*, \rho^{\tilde{i}})$. We also define the operator $Lv(s) = L(v^1(s), \cdots, v^N(s)) = (L^1 v^1(s), \cdots, L^N v^N(s))$. Then $L^i v^i$ is a $|S|$-vector, with $s$th component $L^i v^i(s)$.*

---

For discrete $S$ and bounded $r^i$, it follows from Lemma 5.6.1 in Puterman (2014) that $L^i v^i \in V$ for all $v^i \in V$. Therefore $Lv \in \mathbb{V}$ for all $v \in \mathbb{V}$. And in this paper, we consider the following assumptions in Markov games with state perturbation adversaries.

---

**Assumption 4.4.**

*(1) Bounded rewards; $|r^i(s, a, b)| \leq M^i < M < \infty$ for all $i \in \mathcal{N}$, $a \in A$, $b \in B$, and $s \in S$.*

*(2) Finite state and action spaces: all $S, A^i, B^{\tilde{i}}$ are finite.*

*(3) Stationary transition probability and reward functions.*

*(4) $f(s, \cdot)$ is a bijection for any fixed $s \in S$.*

*(5) All agents share one common reward function.*

---

Finite state and action spaces, bounded rewards, stationary transition kernels, and stationary reward functions are common assumptions in both reinforcement learning and multi-agent reinforcement learning literature (Puterman, 2014; Başar & Olsder, 1998). Additionally, the bijection property of perturbation functions implies that in a finite MG-SPA, adversaries that adopt deterministic policies provide a permutation on the state space. Collaboration and coordination among agents are often required in real-world scenarios to achieve a common goal. In such cases, a shared reward function can motivate agents to work together effectively. Moreover, the assumption of a shared reward function is necessary to transform an MG-SPA into a zero-sum two-agent extensive-form game in our proof. Although these assumptions do not always hold true in real-world applications, they provide good properties for an MG-SPA and enable the first attempt of theoretical analysis on an MG-SPA. The next two propositions characterize the properties of the minimax operator $L$ and space $\mathbb{V}$. We provide the proof in Appendix A.2. These contraction mapping and complete space results are used in the proof of RE existence for an MG-SPA.

---

**Proposition 4.5** (Contraction mapping)**.**

*Suppose $0 \leq \gamma < 1$, and Assumption 4.4 holds. Then $L$ is a contraction mapping on $\mathbb{V}$.*

---

> **Proposition 4.6** (Complete Space). *The space $\mathbb{V}$ is a complete normed linear space.*

In Theorem 4.7, we show some fundamental characteristics of an MG-SPA. In (1), we show that an optimal value function of an MG-SPA satisfies the Bellman equations by applying the Squeeze theorem [Theorem 3.3.6, Sohrab (2003)]. Theorem 4.7-(2) shows that the unique solution of the Bellman equation exists, a consequence of the fixed-point theorem (Smart, 1980). Therefore, the optimal value function of an MG-SPA exists under Assumption 4.4. By introducing (3), we characterize the relationship between the optimal value function and a robust equilibrium. However, (3) does not imply the existence of an RE. To this end, in (4), we formally establish the existence of RE when the optimal value function exists. We formulate a $2N$-player Extensive-form game (EFG) (Osborne & Rubinstein, 1994; Von Neumann & Morgenstern, 2007) based on the optimal value function such that its Nash equilibrium (NE) policy is equivalent to an RE policy of the MG-SPA.

> **Theorem 4.7.**
>
> *Suppose $0 \le \gamma < 1$ and Assumption 4.4 holds.*
>
> *(1) (Solution of Bellman equation) A value function $v_* \in \mathbb{V}$ is an optimal value function if for all $i \in \mathcal{N}$, the point-wise value function $v_*^i \in V$ satisfies the corresponding Bellman Equation (6), i.e. $v_* = Lv_*$.*
>
> *(2) (Existence and uniqueness of optimal value function) There exists a unique $v_* \in \mathbb{V}$ satisfying $Lv_* = v_*$, i.e. for all $i \in \mathcal{N}$, $L^i v_*^i = v_*^i$.*
>
> *(3) (Robust equilibrium and optimal value function) A joint policy $d_* = (\pi_*, \rho_*)$, where $\pi_* = (\pi_*^1, \cdots, \pi_*^N)$ and $\rho_* = (\rho_*^{\tilde{1}}, \cdots, \rho_*^{\tilde{N}})$, is a robust equilibrium if and only if $v^{d_*}$ is the optimal value function.*
>
> *(4) (Existence of robust equilibrium) There exists a mixed RE for an MG-SPA.*

*Proof.* The full proof of Theorem 4.7 is presented in Appendix A, specifically in A.3. We provide a high-level proof sketch here. Our proof consists of two main parts: 1. Constructing an extensive-form game that is connected to an MG-SPA. 2. Proof of Theorem 4.7. In the first part, we begin by constructing an extensive-form game (EFG) whose payoff function is related to the value functions of an MG-SPA (Appendix A.1). Using the EFG as a tool, we can analyze the properties of an MG-SPA. We provide insights into solving an MG-SPA by solving a constructed EFG: a robust equilibrium (RE) of an MG-SPA can be derived from a Nash equilibrium of an EFG when the EFG's payoff function is related to the optimal value function of the MG-SPA (Lemma A.7 in Appendix). Thus, by providing conditions under which a Nash equilibrium of a well-constructed EFG exists (Appendix A.1.1), we can prove the existence of an RE of the MG-SPA (Theorem 4.7-(4)). The existence of an optimal value function is not yet proven and is left for the second part. In the second part, we prove Theorem 4.7-(1) by showing that for all $i$, there exists a $v^i \in V$ such that $v^i \ge Lv^i$, then $v^i \ge v_*^i$, and there also exists a $v^i \in V$ such that $v^i \le Lv^i$, then $v^i \le v_*^i$. Propositions 4.5 and 4.6 enable us to use Banach Fixed-Point Theorem (Smart, 1980) to prove Theorem 4.7-(2). The proof of Theorem 4.7-(3) benefits from the definitions of the optimal value function and robust equilibrium, Theorem 4.7-(1) and (2). Finally, given the existence of the optimal value function and the results from the first part, we prove the existence of an RE. $\qquad\square$

Though the existence of NE in a stochastic game with perfect information has been investigated (Shapley, 1953; Fink, 1964), it is still an open and challenging problem when players have partially observable information (Hansen et al., 2004; Yang & Wang, 2020a). There is a bunch of literature developing algorithms trying to find the NE in Dec-POMDP or partially observable stochastic game (POSG), and conducting algorithm analysis assuming that NE exists (Chades et al., 2002; Hansen et al., 2004; Nair et al., 2002) without proving the conditions for the existence of NE. Once established the existence of RE, we design algorithms to find it. In Section 5, we first develop a robust multi-agent Q-learning (RMAQ) algorithm with a convergence guarantee, then propose a robust multi-agent actor-critic (RMAAC) algorithm to handle the case with high-dimensional state-action spaces.

---

**Remark 4.8** (Heterogeneous agents and adversaries). *In the above problem formulation, we assume all agents have the same type of state perturbations (share one $f$), and all adversaries have the same level of perturbation power (share one $\epsilon$), which made the notation more concise and the analysis more tractable. However, these assumptions are sometimes unrealistic in practice since agents/adversaries may have different capabilities. To introduce heterogeneous agents and adversaries to an MG-SPA, we let each agent $i$ has its own perturbation function $f^i$, and each adversary $\tilde{i}$ has its own perturbation power constraint $\epsilon^i$, such that $\tilde{s}^i = f^i(s, b^i) \in \mathcal{B}(\epsilon^i, s)$. We use $f(s, b) = (f^1(s, b^1), \cdots, f^N(s, b^N)) = \tilde{s}$ to denote the joint perturbation function. When all perturbation functions $f^i(s, \cdot)$ are bijective for any fixed $s \in S$, the joint perturbation function $f(s, \cdot)$ is also a bijection. Assumption 4.4-(4) still holds. When constructing an extensive-form game, the action set for player P1 is defined as $\tilde{S} = \mathcal{B}^1(\epsilon^1, s) \times \mathcal{B}^2(\epsilon^2, s) \times \cdots \mathcal{B}^N(\epsilon^N, s)$ instead of $\tilde{S} = \mathcal{B}(\epsilon, s) \times \cdots \times \mathcal{B}(\epsilon, s)$. The subsequent proofs still hold and they are not affected by the introduction of heterogeneous agents and adversaries. After extending our MG-SPA framework to handle heterogeneous agents and adversaries, we can model more complex and realistic multi-agent systems.*

---

**Remark 4.9** (A reduced case of MG-SPA: a single-agent system). *When there is only one agent in the system, the MG-SPA problem reduces to a single-agent robust RL problem with state uncertainty, which has been studied in the literature (Zhang et al., 2020a; 2021). In this case, single-agent robust RL with state uncertainty can be seen as a specific and special instance of MG-SPA presented in this paper. However, the proposed analysis and algorithm in this paper provide a new perspective and approach to single-agent robust reinforcement learning, by explicitly modeling the adversary's perturbations and optimizing the agent's policy against them in a game-theoretic framework. Moreover, the presence of multiple agents and adversaries in the system can result in more complex and challenging interactions, joint actions and policies that do not present in single-agent RL problems. Our proposed MG-SPA framework allows for the modeling of a wide range of agent interactions, including cooperative, competitive, and mixed interactions.*

---

### 4.3 History-dependent-policy-based Robust Equilibrium

It is natural and desirable to consider history-dependent policies in robust MARL with state perturbations, since the agents may not fully capture the state uncertainty from the current state information, and a policy that only depends on the current state may not be sufficient for ensuring robustness. The history-dependent policy allows agents to take into account past observations when making decisions, which helps agents better reason about the adversaries' possible strategies and intentions. This is particularly true in the case of Dec-POMDPs and POSGs, where the agent cannot fully observe the state. Therefore, we further extend the above Markov-policy-based RE to a history-dependent-policy-based robust equilibrium and discuss Theorem 4.7 under history-dependent policies in this subsection. In Section 5, we also discuss how the proposed algorithms can adapt to historical state input. We further validate that a history-dependent-policy-based RE outperforms a Markov-policy-based RE in Section 6.

In this subsection, we clarify the generalization steps of extending Markov-policy-based RE to history-dependent-policy-based RE. We first introduce the definition of history-dependent policy with a finite time horizon $h$ in an MG-SPA. We then give the formal definition of a history-dependent-policy-based robust equilibrium. Finally, we show that Theorem 4.7 still holds when agents and adversaries adopt history-dependent policies.

We consider an MG-SPA with a time horizon $h$, in which adversaries and agents respectively observe the states and perturbed states in the latest $h$ time steps and adopt history-dependent policies. More concretely, adversary $\tilde{i}$ can manipulate the corresponding agent $i$'s state at time $t$ by using a history-dependent policy $\rho_h^{\tilde{i}}(\cdot|s_{h,t})$ and agent $i$ chooses its actions using a history-dependent policy $\pi_h^i(\cdot|\tilde{s}_{h,t}^i)$. Specifically, once adversary $\tilde{i}$ gets the true state $s_t$ at time $t$, it chooses an action $b_t^{\tilde{i}}$ according to a history-dependent policy $\rho_h^{\tilde{i}} : S_h \to \Delta(B^{\tilde{i}})$, where $s_{h,t} = (s_t, \cdots, s_{t-h+1}) \in S_h$ is a concatenated state consists of the latest $h$ time steps of states. According to a perturbation function $f$, adversary $\tilde{i}$ perturbs state $s_t$ to $\tilde{s}_t^i = f(s_t, b_t^{\tilde{i}}) \in S$. The adversaries' joint policy is defined as $\rho_h(b|s_h) = \prod_{\tilde{i} \in \mathcal{M}} \rho_h^{\tilde{i}}(b^{\tilde{i}}|s_h)$. Agent $i$ chooses its action $a_t^i$ for $\tilde{s}_{h,t}^i = (\tilde{s}_t^i, \cdots, \tilde{s}_{t-h+1}^i) \in S_h$ with probability $\pi_h^i(a_t^i|\tilde{s}_{h,t}^i)$ according to a history dependent policy $\pi_h^i : S_h \to \Delta(A^i)$. The agents' joint policy

is defined as $\pi_h(a|\tilde{s}_h) = \prod_{i \in \mathcal{N}} \pi_h^i(a^i|\tilde{s}_h^i)$. Then a joint history-dependent policy $d_{h,*} = (\pi_{h,*}, \rho_{h,*})$ where $\pi_{h,*} = (\pi_{h,*}^1, \cdots, \pi_{h,*}^N)$ and $\rho_{h,*} = (\rho_{h,*}^{\tilde{1}}, \cdots, \rho_{h,*}^{\tilde{N}})$ is said to be in a history-dependent-policy-based robust equilibrium if and only if, for any $i \in \mathcal{N}, \tilde{i} \in \mathcal{M}, s \in S$,

$$v^{(\pi_{h,*}^{-i}, \pi_{h,*}^i, \rho_{h,*}^{-\tilde{i}}, \rho_h^{\tilde{i}}), i}(s) \geq v^{(\pi_{h,*}^{-i}, \pi_{h,*}^i, \rho_{h,*}^{-\tilde{i}}, \rho_{h,*}^{\tilde{i}}), i}(s) \geq v^{(\pi_{h,*}^{-i}, \pi_h^i, \rho_{h,*}^{-\tilde{i}}, \rho_{h,*}^{\tilde{i}}), i}(s).$$

It is worth noting that the main differences between history-dependent-policy-based RE and Markov-policy-based RE are the definition and notation of policies and states. A Markov-policy-based RE is a special case of a history-dependent-policy-based RE by adopting the time horizon $h = 1$. We also notice that these two REs' definitions are the same if we remove the subscript $h$ from the concatenated state and history-dependent policies. Therefore, in this paper, we use notations without the time horizon subscripts, i.e. Markov policy and Markov-policy-based RE, to avoid redundant and complicated notations. While in this subsection, we clarify the definitions of history-dependent policy and history-dependent-policy-based RE and show that Theorem 4.7 still holds when agents and adversaries use history-dependent policies in the following corollary.

> **Corollary 4.9.1.** *Theorem 4.7 still holds when all agents and adversaries in an MG-SPA use history-dependent policies with a finite time horizon.*

*Proof.* See Appendix A.4. $\qquad\square$

> **Remark 4.10** (MG-SPA, Dec-POMDP, and POSG)**.** *Decentralized Partially Observable Markov Decision Process (Dec-POMDP) enables a team of agents to optimize policies with partial observable states (Oliehoek et al., 2016; Nair et al., 2002), while a Partially Observable Stochastic Game (POSG) (Hansen et al., 2004; Emery-Montemerlo et al., 2004) is an extension of stochastic games with imperfect information that can handle partial observable states. We are inspired by them to consider history-dependent policies for our proposed MG-SPA problem. However, there are several differences between MG-SPA, Dec-POMDP and POSG. First, unlike Dec-POMDP, an MG-SPA does not restrict all agents to share the same interest or reward function. The proposed MG-SPA framework is applicable for modeling different relationships between agents, including cooperative, competitive, or mixed interactions. Second, neither Dec-POMDP nor POSG considers the worst-case state perturbation scenarios. In contrast, in an MG-SPA, state perturbation adversaries receive opposite rewards to the agents, which motivates them to find the worst-case state perturbations to minimize the agents' returns. As we explained in the introduction, considering worst-case state perturbations is important for MARL. Third, while in a Dec-POMDP or a POSG, all agents cannot observe the true state information, in an MG-SPA, adversaries can access the true state and utilize it to select state perturbation actions. Based on these differences in problem formulation, Dec-POMDP and POSG methods cannot solve the proposed MG-SPA problem.*

## 5 Algorithm

### 5.1 Robust Multi-Agent Q-learning (RMAQ) Algorithm

By solving the Bellman equation, we are able to get the optimal value function of an MG-SPA as shown in Theorem 4.7. We therefore develop a value iteration (VI)-based method called robust multi-agent Q-learning (RMAQ) algorithm. Recall the Bellman equation using action-value function in (5), the optimal action-value $q_*$ satisfies $q_*^i(s, a, b) := r^i(s, a, b) + \gamma \mathbb{E}\left[\sum_{s' \in S} p(s'|s, a, b) q_*^i(s', a', b')|a' \sim \pi_*(\cdot|\tilde{s}'), b' \sim \rho_*(\cdot|s')\right]$. As a consequence, the tabular-setting RMAQ update can be written as below,

$$q_{t+1}^i(s_t, a_t, b_t) = (1 - \alpha_t) q_t^i(s_t, a_t, b_t) + \tag{7}$$

$$\alpha_t \left[ r_t^i + \gamma \sum_{a_{t+1} \in A} \sum_{b_{t+1} \in B} \pi_{*,t}^{q_t}(a_{t+1}|\tilde{s}_{t+1}) \rho_{*,t}^{q_t}(b_{t+1}|s_{t+1}) q_t^i(s_{t+1}, a_{t+1}, b_{t+1}) \right],$$

where $(\pi_{*,t}^{q_t}, \rho_{*,t}^{q_t})$ is an NE policy by solving the $2N$-player extensive-form game (EFG) based on a payoff function $(q_t^1, \cdots, q_t^N, -q_t^1, \cdots, -q_t^N)$. The joint policy $(\pi_{*,t}^{q_t}, \rho_{*,t}^{q_t})$ is used in updating $q_t$. All related definitions

of the EFG $(q_t^1, \cdots, q_t^N, -q_t^1, \cdots, -q_t^N)$ are introduced in Appendix A.1. How to solve an EFG is out of the scope of this work, algorithms to do this exist in the literature (Čermák et al., 2017; Kroer et al., 2020). Note that, in RMAQ, each agent's policy is related to not only its own value function, but also other agents' value function. This *multi-dependency* structure considers the interactions between agents in a game, which is different from the Q-learning in single-agent RL that considers optimizing its own value function. Meanwhile, establishing the convergence of a multi-agent Q-learning algorithm is also a general challenge. Therefore, we try to establish the convergence of (7) in Theorem 5.2, motivated from Hu & Wellman (2003). Due to space limitation, in Appendix B.1, we prove that RMAQ is guaranteed to get the optimal value function $q_* = (q_*^1, \cdots, q_*^N)$ by updating $q_t = (q_t^1, \cdots, q_t^N)$ recursively using (7) under Assumptions 5.1.

> **Assumption 5.1.**
>
> *(1) State and action pairs have been visited infinitely often. (2) The learning rate $\alpha_t$ satisfies the following conditions: $0 \leq \alpha_t < 1$, $\sum_{t \geq 0} \alpha_t^2 \leq \infty$; if $(s, a, b) \neq (s_t, a_t, b_t)$, $\alpha_t(s, a, b) = 0$. (3) An NE of the $2N$-player EFG based on $(q_t^1, \cdots, q_t^N, -q_t^1, \cdots, -q_t^N)$ exists at each iteration $t$.*

> **Theorem 5.2.**
>
> *Under Assumption 5.1, the sequence $\{q_t\}$ obtained from (7) converges to $\{q_*\}$ with probability 1, which are the optimal action-value functions that satisfy Bellman equations (5) for all $i = 1, \cdots, N$.*

Assumption 5.1-(1) is a typical ergodicity assumption used in the convergence analysis of Q-learning (Littman & Szepesvári, 1996; Hu & Wellman, 2003; Szepesvári & Littman, 1999; Qu & Wierman, 2020; Sutton & Barto, 1998). And for Q-learning algorithm design papers that the exploration property is not the main focus, this assumption is also a common assumption (Fujimoto et al., 2019). For exploration strategies in RL (McFarlane, 2018), researchers use $\epsilon$-greedy exploration (Gomes & Kowalczyk, 2009), UCB (Jin et al., 2018; Azar et al., 2017), Thompson sampling (Russo et al., 2018), Boltzmann exploration (Cesa-Bianchi et al., 2017), etc. For assumption 5.1-(3) in multi-agent Q-learning, researchers have found that the convergence is not necessarily so sensitive to the existence of NE for the stage games during training (Hu & Wellman, 2003; Yang et al., 2018). In particular, under Assumption 4.4, an NE of the $2N$-player EFG exists, which has been proved in Lemma A.6 in Appendix A.1. We also provide an example in the experiment part (the two-player game) where assumptions are indeed satisfied, and our RMAQ algorithm successfully converges to an RE of the corresponding MG-SPA.

## 5.2 Robust Multi-Agent Actor-Critic (RMAAC) Algorithm

According to the above descriptions of a tabular RMAQ algorithm, each learning agent has to maintain $N$ action-value functions. The total space requirement is $N|S||A|^N|B|^N$ if $|A^1| = \cdots = |A^N|, |B^1| = \cdots = |B^N|$. This space complexity is linear in the number of joint states, polynomial in the number of agents' joint actions and adversaries' joint actions, and exponential in the number of agents. The computational complexity is mainly related to algorithms to solve an extensive-form game (Čermák et al., 2017; Kroer et al., 2020). However, even for general-sum normal-form games, computing an NE is known to be PPAD-complete, which is still considered difficult in game theory literature (Daskalakis et al., 2009; Chen et al., 2009; Etessami & Yannakakis, 2010). These properties of the RMAQ algorithm motivate us to develop an actor-critic method to handle high-dimensional space-action spaces, which can incorporate function approximation into the update (Konda & Tsitsiklis, 1999).

We consider each agent $i$'s policy $\pi^i$ is parameterized as $\pi_{\theta^i}$ for $i \in \mathcal{N}$, and the adversary's policy $\rho^{\tilde{i}}$ is parameterized as $\rho_{\omega^i}$. We denote $\theta = (\theta^1, \cdots, \theta^N)$ as the concatenation of all agents' policy parameters, $\omega$ has the similar definition. For simplicity, we omit the subscript $\theta_i, \omega_i$, since the parameters can be identified by the names of policies. Then the value function $v^i(s)$ under policy $(\pi, \rho)$ satisfies

$$v^{\pi,\rho,i}(s) = \mathbb{E}_{a \sim \pi, b \sim \rho} \left[ \sum_{s' \in S} p(s'|s, a, b)[r^i(s, a, b) + \gamma v^{\pi,\rho,i}(s')] \right]. \tag{8}$$

We establish the general policy gradient with respect to the parameter $\theta, \omega$ in the following theorem. Then we propose our robust multi-agent actor-critic algorithm (RMAAC) which adopts a centralized-training decentralized-execution algorithm structure in MARL literature (Lowe et al., 2017; Foerster et al., 2018).

---

**Theorem 5.3** (Policy Gradient in RMAAC for MG-SPA). *For each agent $i \in \mathcal{N}$ and adversary $\tilde{i} \in \mathcal{M}$, the policy gradients of the objective $J^i(\theta, \omega)$ with respect to the parameter $\theta, \omega$ are:*

$$\nabla_{\theta^i} J^i(\theta, \omega) = \mathbb{E}_{(s,a,b) \sim p(\pi, \rho)} \left[ q^{i,\pi,\rho}(s, a, b) \nabla_{\theta^i} \log \pi^i(a^i | \tilde{s}^i) \right] \tag{9}$$

$$\nabla_{\omega^i} J^i(\theta, \omega) = \mathbb{E}_{(s,a,b) \sim p(\pi, \rho)} \left[ q^{i,\pi,\rho}(s, a, b) [\nabla_{\omega^i} \log \rho^{\tilde{i}}(b^{\tilde{i}} | s) + reg] \right] \tag{10}$$

*where $reg = \nabla_{\tilde{s}^i} \log \pi^i(a^i | \tilde{s}^i) \nabla_{b^{\tilde{i}}} f(s, b^{\tilde{i}}) \nabla_{\omega^i} \rho^{\tilde{i}}(b^{\tilde{i}} | s)$.*

---

*Proof.* See details in Appendix B.2.1. □

We put the pseudo-code of RMAAC in Appendix B.2.3.

---

**Remark 5.4** (History-dependent Policy). *RMAAC can calculate history-dependent policies by using recent observations as the policy input. For example, DQN (Mnih et al., 2015) maps history–action pairs to scalar estimates of Q-value. It uses the history (4 most recent frames) of the states and the action as the inputs of the neural network.*

---

# 6 Experiment

We aim to answer the following questions through experiments: (1) Can RMAQ algorithm find an RE? (2) Are RE policies robust to state uncertainties? (3) Does RMAAC algorithm outperform other MARL and robust MARL algorithms in terms of robustness? The host machine used in our experiments is a server configured with AMD Ryzen Threadripper 2990WX 32-core processors and four Quadro RTX 6000 GPUs. All experiments are performed on Python 3.5.4, Gym 0.10.5, Numpy 1.14.5, Tensorflow 1.8.0, and CUDA 9.0. Our code is public on `https://github.com/sihongho/robust_marl_with_state_uncertainty`.

## 6.1 Robust Multi-Agent Q-learning (RMAQ)

We show the performance of the proposed RMAQ algorithm by applying it to a two-player game. We first introduce the designed two-player game. Then, to answer the first and second questions, we investigate the convergence of this algorithm and compare the performance of robust equilibrium policies with other agents' policies under different adversaries' policies.

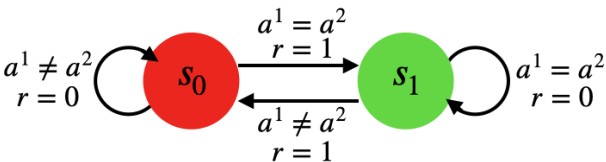

Figure 3: Two-player game: each player has two states and the same action set with size 2. Under state $s_0$, two players get the same reward 1 when they choose the same action. At state $s_1$, two players get the same reward 1 when they choose different actions. One state switches to another state only when two players get a reward.

**Two-player game:** For the game in Figure 3, two players have the same action space $A = \{0, 1\}$ and state space $S = \{s_0, s_1\}$. The two players get the same positive rewards when they choose the same action under state $s_0$ or choose different actions under state $s_1$. The state does not change until these two players get a positive reward. Possible Nash equilibrium (NE) in this game can be $\pi_1^* = (\pi_1^1, \pi_1^2)$ that player 1 always chooses action 1, player 2 chooses action 1 under state $s_0$ and action 0 under state $s_1$; or $\pi_2^* = (\pi_2^1, \pi_2^2)$ that player 1 always chooses action 0, player 2 chooses action 0 under state $s_0$ and action 1 under state $s_1$. When using the NE policy, these two players always get the same positive rewards. The optimal discounted state value of this game is

$v_*^i(s) = 1/(1 - \gamma)$ for all $s \in S, i \in \{1, 2\}$, $\gamma$ is the reward discounted rate. We set $\gamma = 0.99$, then $v_*^i(s) = 100$.

**MG-SPA formulation for the two-player game:** According to the definition of MG-SPA, we add two adversaries, one for each player to perturb the state and get a negative reward of the player. They have the same action space $B = \{0, 1\}$, where 0 means do not disturb, 1 means perturb the observed state to another one. Sometimes no perturbation would be a good choice for adversaries. For example, when the true state is $s_0$, players are using $\pi_1^*$, if adversary 1 does not perturb player 1's observation, player 1 will still select action 1. While adversary 2 changes player 2's observation to state $s_1$, player 2 will choose action 0 which is not the same as player 1's action 1. Thus, players always fail the game and get no rewards. A robust equilibrium for MG-SPA would be $\tilde{d}_* = (\tilde{\pi}_*^1, \tilde{\pi}_*^2, \tilde{\rho}_*^1, \tilde{\rho}_*^2)$ that each player chooses actions with equal probability and so do adversaries. The optimal discounted state value of corresponding MG-SPA is $\tilde{v}_*^i(s) = 1/2(1 - \gamma)$ for all $s \in S, i \in \{1, 2\}$ when players use robust equilibrium (RE) policies. We use $\gamma = 0.99$, then $\tilde{v}_*^i(s) = 50$. For more explanations of this two-player game and corresponding MG-SPA formulation, please see Appendix C.1.

**Implementing RMAQ on the two-player game:** We initialize $q^1(s, a, b) = q^2(s, a, b) = 0$ for all $s, a, b$. After observing the current state, adversaries choose their actions to perturb the agents' state. Then players execute their actions based on the perturbed state information. They then observe the next state and rewards. Then every agent updates its $q$ according to (7). In the next state, all agents repeat the process above. The training stops after 7500 steps. When updating the Q-values, the agent applies a NE policy from the Extensive-form game based on $(q^1, q^2, -q^1, -q^2)$.

**Training results:** After 7000 steps of training, we find that agents' Q-values stabilize at certain values. Since the dimension of $q$ is a bit high as $q \in \mathbb{R}^{32}$, we compare the optimal state value $\tilde{v}_*$ and the total discounted rewards in Table 1. The value of the total discounted reward converges to the optimal state value of the corresponding MG-SPA. This two-player game experiment result validates the convergence of our RMAQ method and the answer to the first question is 'Yes'.

**Testing results:** We further test well-trained RE policy when 'strong' adversaries exist. 'Strong' adversary means its probability of modifying players' observations is larger than the probability of no perturbations in the state information. We make two players play the game using 3 different policies for 1000 steps under different adversaries. The accumulated rewards and total discounted rewards are calculated. We use the robust equilibrium (of the MG-SPA), the Nash equilibrium (of the original game), and a baseline policy (two players use deterministic policies) and report the result in Figure 4. The vertical axis is the accumulated/discounted reward, and the horizon axis is the probability that the adversary will attack/perturb the state. And we let these two adversaries share the same policy. We can see as the probability increase, the accumulated and discounted rewards of RE players are stable but those rewards of NE players and baseline players keep decreasing. These experimental results show that the RE policy is robust to state uncertainties. It turns out the answer to the second question is 'Yes' as well.

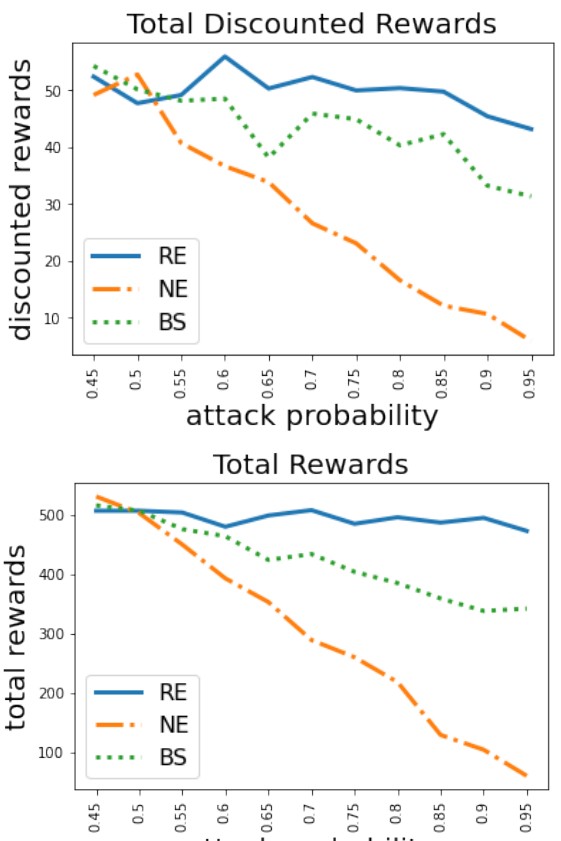

Figure 4: RE policy outperforms other policies in terms of total discounted rewards and total accumulated rewards when strong state uncertainties exist.

**Discussion:** Even for general-sum normal-form games, computing an NE is known to be PPAD-complete, which is still considered difficult in game theory literature (Conitzer & Sandholm, 2002; Etessami & Yannakakis,

Table 1: Convergence Values of Total Discounted Rewards when Training Ends

|  | $v^1(s_0)$ | $v^2(s_0)$ | $v^1(s_1)$ | $v^2(s_1)$ | $\tilde{v}^1_*(s_0)$ | $\tilde{v}^2_*(s_0)$ | $\tilde{v}^1_*(s_1)$ | $\tilde{v}^2_*(s_1)$ |
|---|---|---|---|---|---|---|---|---|
| value | 49.99 | 49.99 | 49.99 | 49.99 | 50.00 | 50.00 | 50.00 | 50.00 |

2010). Therefore, we do not anticipate that the RMAQ algorithm can scale to very large MARL problems. In the next subsection, we show RMAAC with function approximation can handle large-scale MARL problems.

## 6.2 Robust Multi-Agent Actor-Critic (RMAAC)

To answer the third question, we compare our RMAAC algorithm with two benchmark MARL algorithms: MADDPG (`https://github.com/openai/maddpg`) (Lowe et al., 2017) which does not consider robustness, and M3DDPG (`https://github.com/dadadidodi/m3ddpg`) (Li et al., 2019), a robust MARL algorithm which considers uncertainties from opponents' policies altering. M3DDPG utilizes adversarial learning to train robust policies. We run experiments in several benchmark multi-agent scenarios, based on the multi-agent particle environments (MPE) (Lowe et al., 2017). The hyper-parameters used to train RMAAC and the baseline algorithms are summarized in Appendix C.2.2, Table 4.

**Experiment procedure:** We first train agents' policies using RMAAC, MADDPG and M3DDPG, respectively. For our RMAAC algorithm, we set the constraint parameter $\epsilon = 0.5$. And we choose two types of perturbation functions to validate the robustness of trained policies under different MG-SPA models. The first one is the linear noise format that $f_1(s, b^{\tilde{i}}) := s + b^{\tilde{i}}$, i.e. the perturbed state $\tilde{s}^i$ is calculated by adding a random noise $b^{\tilde{i}}$ generated by adversary $\tilde{i}$ to the true state $s$. And $f_2(s, b^{\tilde{i}}) := s + Gaussian(b^{\tilde{i}}, \Sigma)$, where the adversary $\tilde{i}$'s action $b^{\tilde{i}}$ is the mean of the Gaussian distribution. And $\Sigma$ is the covariance, we set it as $I$, i.e. an identity matrix. We call it Gaussian noise format. These two formats $f_1, f_2$ are commonly used in adversarial training (Creswell et al., 2018; Zhang et al., 2020a; 2021). Then we test the well-trained policies in the optimally disturbed environment (injected noise is produced by those adversaries trained with RMAAC algorithm). The testing step is chosen as 10000 and each episode contains 25 steps. All hyperparameters used in experiments for RMAAC, MADDPG and M3DDPG are attached in Appendix C.2.2. Note that since the rewards are defined as negative values in the used multi-agent environments, we add the same baseline (100) to rewards for making them positive. Then it's easier to observe the testing results and make comparisons. Those used MPE scenarios are Cooperative communication (CC), Cooperative navigation (CN), Physical deception (PD), Predator prey (PP) and Keep away (KA). The first two scenarios are cooperative games, the others are mixed games. To investigate the algorithm performance in more complicated situations, we also run experiments in a scenario with more agents, which is called Predator prey+ (PP+). More details of these games are in Appendix C.2.1.

**Experiment results:** In Figure 5 and Table 2, we report the mean episode testing rewards and variance of 10000 steps testing rewards, respectively. We will use mean rewards and variance for short in the following experimental report and explanations. In the table and figure, we use RM, M3, MA for abbreviations of RMAAC, M3DDPG and MADDPG, respectively. In Figure 5, the left five figures are mean rewards under the linear noise format $f_1$, the right ones are under the Gaussian noise format $f_2$. Under the optimally disturbed environment, agents with RMAAC policies get the highest mean rewards in almost all scenarios no matter what noise format is used. The only exception is in Keep away under linear noise. However, our RMAAC still achieves the highest rewards when testing in Keep away under Gaussian noise. In Figure 7, we show the comparison results in a complicated scenario with a larger number of agents and RMAAC policies are trained

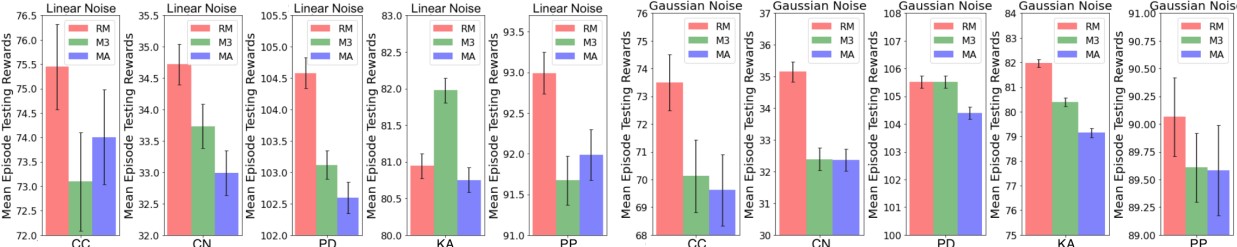

Figure 5: RMAAC outperforms baseline MARL algorithms in terms of mean episode testing rewards under different perturbation functions in most MPE scenarios.

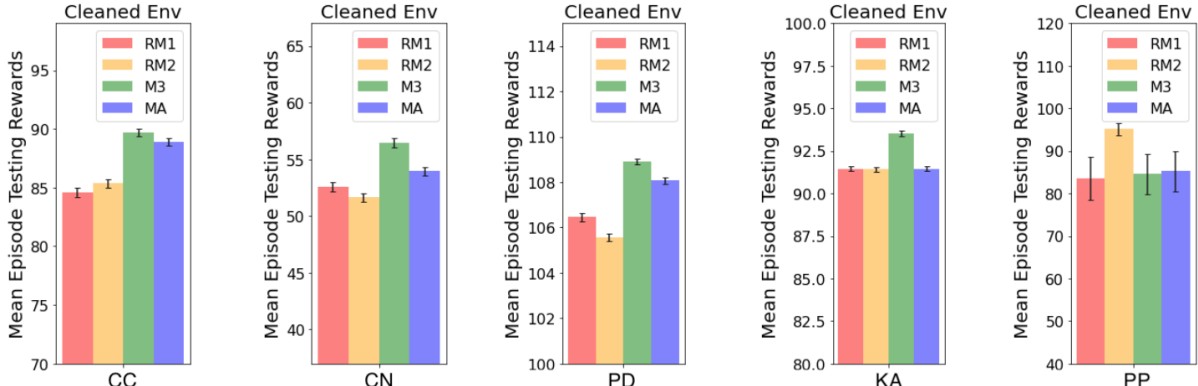

Figure 6: Comparison of mean episode testing rewards using different algorithms and different perturbation functions, under cleaned environments.

with the Gaussian noise format $f_2$. As we can see that the RMAAC policies get the highest reward when testing under optimally perturbed environments, cleaned and randomly perturbed environments. Higher rewards mean agents are performing better. It turns out RMAAC policies outperform the other two baseline algorithms when there exist worst-case state uncertainties. In Table 2, the left three columns report the variance under the linear noise format $f_1$, and the right ones are under the Gaussian noise format $f_2$. RM1 denotes our RMAAC policy trained with the linear noise format $f1$, RM2 denotes our RMAAC policy trained with the Gaussian noise format $f2$. The variance is used to evaluate the stability of the trained policies, i.e. the robustness to system randomness. Because the testing experiments are done in the same environments that are initialized by different random seeds. We can see that, by using our RMAAC algorithm, the agents can get the lowest variance in most scenarios under these two different perturbation formats. Therefore, our RMAAC algorithm is also more robust to the system randomness, compared with the baselines. In summary, our answer to the third question is 'Yes'.

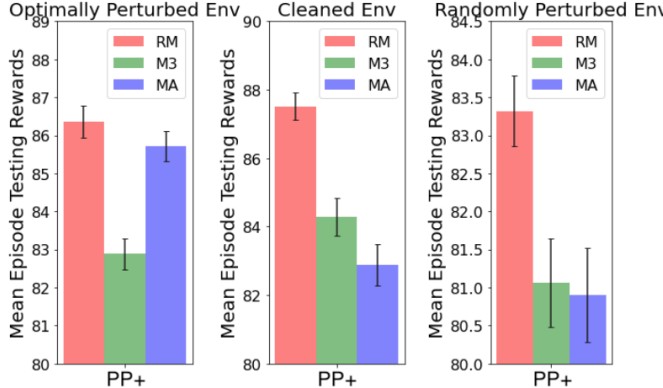

Figure 7: RMAAC outperforms baseline MARL algorithms in terms of mean episode testing rewards in complicated scenarios with a larger number of agents of MPE.

**Interesting results when testing under lighter perturbations and cleaned environments:** We also provide the testing results under a cleaned environment (accurate state information can be attained) and a randomly disturbed environment (injecting standard Gaussian noise into agents' observations). In Figures 6 and 8, we respectively show the comparison of mean episode testing rewards under a cleaned environment and a randomly disturbed environment by using 4 different methods: RM1 denotes our RMAAC policy trained with the linear noise format $f_1$, RM2 denotes our RMAAC policy trained with the Gaussian noise format $f_2$, MA denotes MADDPG, M3 denotes M3DDPG. We can see that only in the Predator prey scenario, our method outperforms others under a cleaned environment. In Figure 8, we can see that our method outperforms others in the Cooperative communication, Keep away and Predator prey scenarios, and achieves similar performance as others in the Cooperative navigation scenario under a randomly perturbed environment.

This kind of performance also happens in robust optimization (Beyer & Sendhoff, 2007; Boyd & Vandenberghe, 2004) and distributionally robust optimization (Delage & Ye, 2010; Rahimian & Mehrotra, 2019; Miao et al., 2021; He et al., 2020; 2023) where the robust solutions outperform other non-robust solutions in the worst-case

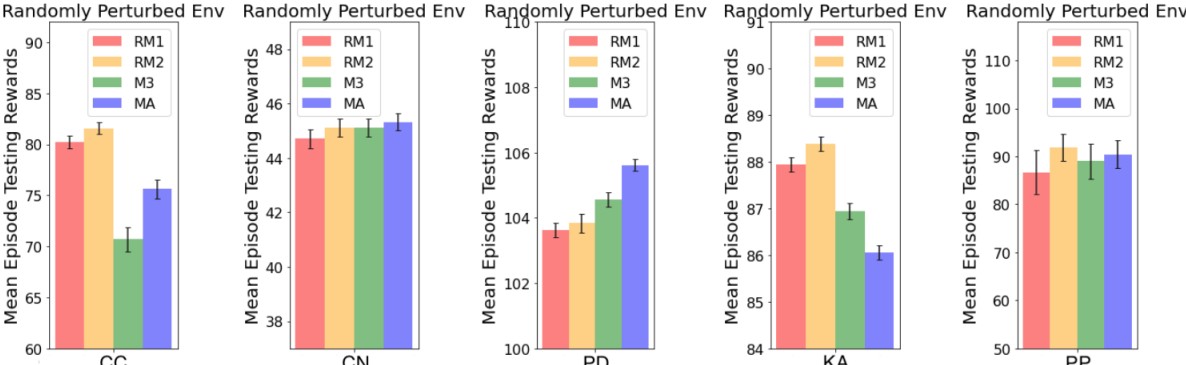

Figure 8: Comparison of mean episode testing rewards using different algorithms and different perturbation functions, under randomly perturbed environments.

Table 2: Variance of Testing Rewards under optimal perturbed environment

| Perturbation function | Linear noise $f_1$ | | | Gaussian noise $f_2$ | | |
|---|---|---|---|---|---|---|
| Algorithms | **RM1** | M3 | MA | **RM2** | M3 | MA |
| Cooperative communication (CC) | **1.007** | 1.311 | 1.292 | **0.872** | 1.012 | 0.976 |
| Cooperative navigation (CN) | **0.322** | 0.357 | 0.351 | **0.322** | 0.349 | 0.359 |
| Physical deception (PD) | 0.225 | 0.218 | **0.217** | 0.244 | **0.161** | 0.252 |
| Keep away (KA) | **0.161** | 0.168 | 0.175 | **0.161** | 0.17 | 0.167 |
| Predator prey (PP) | 3.213 | **0.161** | 3.671 | **2.304** | 2.711 | 2.811 |

scenario. Similarly, for single-agent RL with state perturbations, robust policies perform better compared with baselines under state perturbations (Zhang et al., 2020b). However, there exists a trade-off between optimizing the average performance and the worst-case performance for robust solutions in general, and the robust solutions may get relatively poor performance compared with other non-robust solutions when there is no uncertainty or perturbation in the environment even in a single agent RL problem (Zhang et al., 2020b). Improving the robustness of the trained policy may sacrifice the performance of the decisions when perturbations or uncertainties do not happen. That's why our RMAAC policies only beat all baselines in one scenario when the state uncertainty is eliminated. However, for many real-world systems, we can not assume that agents always have accurate information about the states. Hence, improving the robustness of the policies is very important for MARL as we explained in the introduction. It is worth noting that our RMAAC policies also work well in environments with random perturbations instead of only the worst-case perturbations. As shown in Fig. 8, the performance of our RMAAC policies outperforms the baselines in most scenarios when random noise is introduced into the state.

More experimental results and explanations are provided in Appendix C.2.

**Ablation study:** We conducted ablation studies for RMAAC algorithm. We first study the performance of RMAAC when it is used to train history-dependent policies. Other than using the current information as the input of the policy neural network, we also use history information in the latest three time steps, i.e. $h = 4$. In Table 3, we show the mean and variance of mean episode rewards in 10 runs. We use the same hyper-parameters in training history-dependent policies as training Markov policies. We can see that in

Table 3: Means and Variances of Mean Episode Rewards using Different Polices

| Scenarios | History-dependent policy | Markov policy |
|---|---|---|
| Cooperative communication (CC) | **-52.83** $\pm$ 1.51 | -54.75 $\pm$ 3.03 |
| Cooperative navigation (CN) | **-208.19** $\pm$ 1.68 | -210.41 $\pm$ 1.13 |
| Physical deception (PD) | **7.72** $\pm$ 0.33 | 5.71 $\pm$ 0.19 |
| Keep away (KA) | **-20.69** $\pm$ 0.09 | -21.18 $\pm$ 0.14 |
| Predator prey (PP) | **7.10** $\pm$ 0.17 | 6.116 $\pm$ 0.24 |

all five scenarios, history-dependent policies outperform Markov policies. Besides, we investigate how the robustness performance of RMAAC is affected by varying variances of Gaussian noise format $\Sigma$ and the constraint parameter $\epsilon$. We also investigate the performance of RMAAC under other types of attacks. Please check the experimental setups and results of these ablation studies in Appendix C.3.

## 7 Discussion

Our proposed method provides a foundation for modeling robust agents' interactions in multi-agent reinforcement learning with state uncertainty, and training policies robust to state uncertainties in MARL. However, there are still several urgent and promising problems to solve in this field.

First, exploring heterogeneous agent modeling is an important research direction in robust MARL. Training a team of heterogeneous agents to learn robust control policies presents unique challenges, as agents may have different capabilities, knowledge, and objectives that can lead to conflicts and coordination problems (Lin et al., 2021). State uncertainty can exacerbate the impact of these differences, as agents may not be able to accurately estimate the state of the environment or predict the behavior of other agents. All of these factors make modeling heterogeneous agents in the presence of state uncertainty a challenging problem.

Second, investigating methods for handling continuous state and action spaces can benefit both general MARL problems and our proposed method. While discretization is a commonly used approach for dealing with continuous spaces, it is not always an optimal method for handling high-dimensional continuous spaces, especially when state uncertainty is present. Adversarial state perturbation may disrupt the continuity on the continuous state space, which can lead to difficulties in finding globally optimal solutions using general discrete methods. This is because continuous spaces have infinite possible values, and discretization methods may not be able to accurately represent the underlying continuous structure. When state perturbation occurs, it may lead to more extreme values, which can result in the loss of important information. We will investigate more on methods for continuous state and action space robust MARL in the future.

## 8 Conclusion

We study the problem of multi-agent reinforcement learning with state uncertainties in this work. We model the problem as a Markov game with state perturbation adversaries (MG-SPA), where each agent aims to find out a policy to maximize its own total discounted reward and each associated adversary aims to minimize that. This problem is challenging with little prior work on theoretical analysis or algorithm design. We provide the first attempt at theoretical analysis and algorithm design for MARL under worst-case state uncertainties. We first introduce robust equilibrium as the solution concept for MG-SPA, and prove conditions under which such an equilibrium exists. Then we propose a robust multi-agent Q-learning algorithm (RMAQ) to find such an equilibrium, with convergence guarantees under certain conditions. We also derive the policy gradients and design a robust multi-agent actor-critic (RMAAC) algorithm to handle the more general high-dimensional state-action space MARL problems. We also conduct experiments that validate our methods.

### Acknowledgments

Sihong He, Songyang Han, Sanbao Su and Fei Miao are supported by the National Science Foundation under Grants CNS-1952096, CMMI-1932250, and CNS-2047354 grants. Shaofeng Zou is supported by the National Science Foundation under Grants CCF-2106560, and CCF-2007783.

This material is based upon work supported under the AI Research Institutes program by National Science Foundation and the Institute of Education Sciences, U.S. Department of Education through Award # 2229873 - National AI Institute for Exceptional Education. Any opinions, findings and conclusions or recommendations expressed in this material are those of the author(s) and do not necessarily reflect the views of the National Science Foundation, the Institute of Education Sciences, or the U.S. Department of Education.

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

# Appendix for "Robust Multi-Agent Reinforcement Learning with State Uncertainty"

There are three sections in the appendix: section A for theoretical proof, section B for algorithms, and section C for experiments.

## A   Theory

In this section, we give the full proof of all propositions and theorems in the theoretical analysis of an MG-SPA.

In section A.1, we construct an extensive-form game (EFG) (Başar & Olsder, 1998; Osborne & Rubinstein, 1994; Von Neumann & Morgenstern, 2007) whose payoff function is related to value functions of an MG-SPA. We then give certain conditions under which, a Nash equilibrium for the constructed EFG exists. In section A.2, we prove the propositions 4.5 and 4.6. In section A.3, we give the full proof of Theorem 4.7. In section A.4, we prove Corollary 4.9.1 that Theorem 4.7 applies to history-dependent-policy-based RE as well.

To make the appendix self-contained, we re-show the vector notations and assumptions we have presented in section 4.2. Readers can also **skip** the repeated text and directly go to section A.1.

We follow and extend the vector notations in Puterman (2014). Let $V$ denote the set of bounded real valued functions on $S$ with component-wise partial order and norm $\|v^i\| := \sup_{s \in S} |v^i(s)|$. Let $V_M$ denote the subspace of $V$ of Borel measurable functions. For discrete state space, all real-valued functions are measurable so that $V = V_M$. But when $S$ is a continuum, $V_M$ is a proper subset of $V$. Let $v = (v^1, \cdots, v^N) \in \mathbb{V}$ be the set of bounded real valued functions on $S \times \cdots \times S$, i.e. the across product of $N$ state set and norm $\|v\| := \sup_j \|v^j\|$. We also define the set $Q$ and $\mathbb{Q}$ in a similar style such that $q^i \in Q, q \in \mathbb{Q}$.

For discrete $S$, let $|S|$ denote the number of elements in $S$. Let $r^i$ denote a $|S|$-vector, with $s$th component $r^i(s)$ which is the expected reward for agent $i$ under state $s$. And $P$ the $|S| \times |S|$ matrix with $(s, s')$th entry given by $p(s'|s)$. We refer to $r^i_d$ as the reward vector of agent $i$, and $P_d$ as the probability transition matrix corresponding to a joint policy $d = (\pi, \rho)$. $r^i_d + \gamma P_d v^i$ is the expected total one-period discounted reward of agent $i$, obtained using the joint policy $d = (\pi, \rho)$. Let $z$ as a list of joint policy $\{d_1, d_2, \cdots\}$ and $P^0_z = I$, we denote the expected total discounted reward of agent $i$ using $z$ as $v^i_z = \sum_{t=1}^{\infty} \gamma^{t-1} P^{t-1}_z r^i_{d_t} = r^i_{d_1} + \gamma P_{d_1} r^i_{d_2} + \cdots + \gamma^{n-1} P_{d_1} \cdots P_{d_{n-1}} r^i_{t_n} + \cdots$. Now, we define the following minimax operator which is used in the rest of the paper.

---

**Definition A.1** (Minimax Operator, same as definition 4.3).

*For $v^i \in V, s \in S$, we define the nonlinear operator $L^i$ on $v^i(s)$ by $L^i v^i(s) := \max_{\pi^i} \min_{\rho^i} [r^i_d + \gamma P_d v^i](s)$, where $d := (\pi^{-i}_*, \pi^i, \rho^{-\tilde{i}}_*, \rho^{\tilde{i}})$. We also define the operator $Lv(s) = L(v^1(s), \cdots, v^N(s)) = (L^1 v^1(s), \cdots, L^N v^N(s))$. Then $L^i v^i$ is a $|S|$-vector, with $s$th component $L^i v^i(s)$.*

---

For discrete $S$ and bounded $r^i$, it follows from Lemma 5.6.1 in Puterman (2014) that $L^i v^i \in V$ for all $v^i \in V$. Therefore $Lv \in \mathbb{V}$ for all $v \in \mathbb{V}$. And in this paper, we consider the following assumptions in Markov games with state perturbation adversaries.

---

**Assumption A.2** (Same as assumption 4.4).

*(1) Bounded rewards; $|r^i(s, a, b)| \le M^i < M < \infty$ for all $i \in \mathcal{N}$, $a \in A$, $b \in B$ and $s \in S$.*

*(2) Finite state and action spaces: all $S, A^i, B^{\tilde{i}}$ are finite.*

*(3) Stationary transition probability and reward functions.*

*(4) $f(s, \cdot)$ is a bijection for any fixed $s \in S$.*

*(5) All agents share one common reward function.*

---

### A.1 Extensive-form game

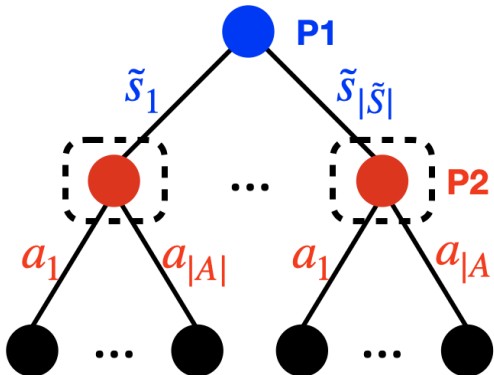

Figure 9: a team extensive-form game

An extensive-form game (EFG) (Başar & Olsder, 1998; Osborne & Rubinstein, 1994; Von Neumann & Morgenstern, 2007) basically involves a tree structure with several nodes and branches, providing an explicit description of the order of players and the information available to each player at the time of his decision.

Look at Figure 9, an EFG involves from the top of the tree to the tip of one of its branches. And a centralized nature player ($P1$) has $|\tilde{S}|$ alternatives (branches) to choose from, whereas a centralized agent ($P2$) has $|A|$ alternatives, and the order of play is that the centralized nature player acts before the centralized agent does. The set $A$ is the same as the agents' joint action set in an MG-SPA, set $\tilde{S}$ is a set of perturbed states constrained by a constrained parameter $\epsilon$. At the end of lower branches, some numbers will be given. These numbers represent the playoffs to the centralized agent (or equivalently, losses incurred to the centralized nature player) if the corresponding paths are selected by the players. We give the formal definition of an EFG we will use in the proof and the main text as follows:

> **Definition A.3.** *An extensive-form game based on $(v^1, \cdots, v^N, -v^1, \cdots, -v^N)$ under $s \in S$ is a finite tree structure with:*
>
> 1. *A player $P1$ has a action set $\tilde{S} = \overbrace{\mathcal{B}(\epsilon, s) \times \cdots \times \mathcal{B}(\epsilon, s)}^{N}$, with a typical element designed as $\tilde{s}$. And $P1$ moves first.*
>
> 2. *Another player $P2$ has an action set $A$, with a typical element designed as $a$. And $P2$ which moves after $P1$.*
>
> 3. *A specific vertex indicating the starting point of the game.*
>
> 4. *A payoff function $g_s(\tilde{s}, a) = (g_s^1(\tilde{s}, a), \cdots, g_s^N(\tilde{s}, a))$ where $g_s^i(\tilde{s}, a) = r^i(s, a, f_s^{-1}(\tilde{s})) + \sum_{s'} p(s'|a, f_s^{-1}(\tilde{s})) v^i(s')$ assigns a real number to each terminal vector of the tree. Player $P1$ gets $-g_s(\tilde{s}, a)$ while player $P2$ gets $g_s(\tilde{s}, a)$.*
>
> 5. *A partition of the nodes of the tree into two player sets (to be denoted by $\bar{N}^1$ and $\bar{N}^2$ for $P1$ and $P2$, respectively).*
>
> 6. *A sub-partition of each player set $\bar{N}^i$ into information set $\{\eta_j^i\}$, such that the same number of immediate branches emanates from every node belonging to the same information set, and no node follows another node in the same information set.*

Note that $f_s(b) := f(s, b) = (f(s, b^{\tilde{1}}), \cdots, f(s, b^{\tilde{N}}))$ is the vector version of the perturbation function $f$ in an MG-SPA. Since in an MG-SPA, $q^i(s, a, b) = r^i(s, a, b) + \sum_{s'} p(s'|s, a, b) v^i(s')$ for all $i = 1, \cdots, N$, $g_s^i(\tilde{s}, a) = q^i(s, a, f_s^{-1}(\tilde{s}))$ as well. We can also use $(q^1, \cdots, q^N, -q^1, \cdots, -q^N)$ to denote an extensive-

form game based on $(v^1, \cdots, v^N, -v^1, \cdots, -v^N)$. Then we define the behavioral strategies for $P1$ and $P2$, respectively in the following definition.

> **Definition A.4.** *(Behavioral strategy) Let $I^i$ denote the class of all information sets of $Pi$, with a typical element designed as $\eta^i$. Let $U^i_{\eta^i}$ denote the set of alternatives of $Pi$ at the nodes belonging to the information set $\eta^i$. Define $U^i = \cup U^i_{\eta^i}$ where the union is over $\eta^i \in I^i$. Let $Y_{\eta^1}$ denote the set of all probability distributions on $U^1_{\eta^1}$, where the latter is the set of all alternatives of $P1$ at the nodes belonging to the information set $\eta^1$. Analogously, let $Z_{\eta^2}$ denote the set of all probability distributions on $U^2_{\eta^2}$. Further define $Y = \cup_{I^1} Y_{\eta^1}, Z = \cup_{I^2} Z_{\eta^2}$. Then, a behavioral strategy $\lambda$ for $P1$ is a mapping from the class of all his information sets $I^1$ into $Y$, assigning one element in $Y$ for each set in $I^1$, such that $\lambda(\eta^1) \in Y_{\eta^1}$ for each $\eta^1 \in I^1$. A typical behavioral strategy $\chi$ for $P2$ is defined, analogously, as a restricted mapping from $I^2$ into $Z$. The set of all behavioral strategies for $Pi$ is called his behavioral strategy set, and it is denoted by $\Gamma^i$.*

The information available to the centralized agent ($P2$) at the time of his play is indicated on the tree diagram in Figure 9 by dotted lines enclosing an area (i.e. the information set) including the relevant nodes. This means the centralized agent is in a position to know exactly how the centralized nature player acts. In this case, a strategy for the centralized agent is a mapping from the collection of his information sets into the set of his actions.

And the behavioral strategy $\lambda$ for $P1$ is a mapping from his information sets and action space into a probability simplex, i.e. $\lambda(\tilde{s}|s)$ is the probability of choosing $\tilde{s}$ given $s$. Similarly, the behavioral strategy $\chi$ for $P2$ is $\chi(a|\tilde{s})$, i.e. the probability of choosing action $a$ when $\tilde{s}$ is given. Note that every behavioral strategy is a mixed strategy. We then give the definition of Nash equilibrium in behavioral strategies for an EFG.

> **Definition A.5.** *(Nash equilibrium in behavioral strategies) A pair of strategies $\{\lambda_* \in \Gamma^1, \chi_* \in \Gamma^2\}$ is said to constitute a Nash equilibrium in behavioral strategies if the following inequalities are satisfied that for all $i = 1, \cdots, N, \lambda \in \Gamma^1, \chi \in \Gamma^2, s \in S$:*
>
> $$J^i(\lambda^i, \lambda_*^{-i}, \chi_*^i, \chi_*^{-i}) \geq J^i(\lambda_*^i, \lambda_*^{-i}, \chi_*^i, \chi_*^{-i}) \geq J^i(\lambda_*^i, \lambda_*^{-i}, \chi^i, \chi_*^{-i}) \qquad (11)$$
>
> *where $J^i(\lambda, \chi)$ is the expected payoff i.e. $\mathbb{E}_{\lambda, \chi}[g_s^i]$ when $P1$ takes $\lambda$, $P2$ takes $\chi$, $\lambda(\tilde{s}|s) = \prod_{i=1}^N \lambda^i(\tilde{s}^i|s)$, $\chi(a|\tilde{s}) = \prod_{i=1}^N \chi^i(a^i|\tilde{s}^i)$.*

In the following parts as well as the main text, when we mention a Nash equilibrium for an EFG, it refers to a Nash equilibrium in behavioral strategies. How to solve an EFG is out of our scope since it has been investigated in much literature (Başar & Olsder, 1998; Schipper, 2017; Slantchev, 2008). And policies $\lambda^i$ and $\chi^i$ can be attained through the marginal probabilities calculation with chain rules (Devore et al., 2012; Mémoli, 2012).

### A.1.1 Existence of NE for an EFG

In Lemma A.6, we give conditions (partial items of Assumption 4.4) under which an NE of the EFG based on $(v^1, \cdots, v^N, -v^1, \cdots, -v^N)$ exists.

> **Lemma A.6.** *Suppose $v^1 = \cdots = v^N$, and $S, A$ are finite. An NE $(\lambda_*, \chi_*)$ of the EFG based on $(v^1, \cdots, v^N, -v^1, \cdots, -v^N)$ exists.*

*Proof.* Since $\tilde{S}$ is a subset of $S$, $\tilde{S}$ is finite when $S$ is finite. When $v^1 = \cdots = v^N$, and $\tilde{S}, A$ are finite, an EFG based on $(v^1, \cdots, v^N, -v^1, \cdots, -v^N)$ degenerates to a zero-sum two-person extensive-form game with finite strategies and perfect recall. Thus, an NE of this EFG exists (Başar & Olsder, 1998; Schipper, 2017; Slantchev, 2008). □

The following Lemma A.7 provides insights into solving an MG-SPA by solving a constructed EFG.

---

**Lemma A.7.** *Suppose $f$ is a bijection when $s$ is fixed and an NE $(\lambda_*, \chi_*)$ exists for an EFG $(v^1, \cdots, v^N, -v^1, \cdots, -v^N)$. We define a joint policy $(\pi_*^v, \rho_*^v)$ as the joint policy implied from the NE $(\lambda_*, \chi_*)$, where $\rho_*^v(b|s) = \lambda_*(\tilde{s} = f_s(b)|s), \pi_*^v(a|\tilde{s} = f_s(b)) = \chi_*(a|\tilde{s})$. Then the joint policy $(\pi_*^v, \rho_*^v)$ satisfies $L^i v^i(s) = r^i_{(\pi_*^v, \rho_*^v)}(s) + \gamma \sum_{s' \in S} p_{(\pi_*^v, \rho_*^v)}(s'|s) v^i(s')$ for all $s \in S$.*

---

*Proof.* The NE of the extensive-form game $(\lambda_*, \chi_*)$ implies that for all $i = 1, \cdots, N, s \in S, \lambda \in \Gamma^1, \chi \in \Gamma^2$, we have

$$J^i(\lambda, \chi_*) \geq J^i(\lambda_*, \chi_*) \geq J^i(\lambda_*, \chi),$$

where $J^i(\lambda, \chi) = \mathbb{E}[r^i(s, a, f_s^{-1}(\tilde{s})) + \sum_{s'} p(s'|s, a, f_s^{-1}(\tilde{s})) v^i(s') | \tilde{s} \sim \lambda(\cdot|s), a \sim \chi(\cdot|\tilde{s})]$ according to Definition A.5. Let $b$ denote $f_s^{-1}(\tilde{s})$, because $f$ is a bijection when $s$ is fixed, $f_s(b) = (f_s(b^{\tilde{1}}), \cdots, f_s(b^{\tilde{N}}))$ is a bijection, and the inverse function $f_s^{-1}(\tilde{s}) = (f_s^{-1}(\tilde{s}^1), \cdots, f_s^{-1}(\tilde{s}^N))$ exists and is a bijection as well, then we have

$$J^i(\lambda_*, \chi_*) = \mathbb{E}\left[r^i(s, a, f_s^{-1}(\tilde{s})) + \sum_{s'} p(s'|s, a, f_s^{-1}(\tilde{s})) v^i(s') | \tilde{s} \sim \lambda_*(\cdot|s), a \sim \chi_*(\cdot|\tilde{s})\right]$$

$$= \mathbb{E}\left[r^i(s, a, b) + \sum_{s'} p(s'|s, a, b) v^i(s') | b \sim \lambda_*(f_s(b)|s), a \sim \chi_*(\cdot|f_s(b))\right]$$

$$= \mathbb{E}\left[r^i(s, a, b) + \sum_{s'} p(s'|s, a, b) v^i(s') | b \sim \rho_*^v(\cdot|s), a \sim \pi_*^v(\cdot|\tilde{s})\right]$$

Similarly, we have

$$J^i(\lambda_*, \chi) = \mathbb{E}\left[r^i(s, a, b) + \sum_{s'} p(s'|s, a, b) v^i(s') | b \sim \rho_*^v(\cdot|s), a \sim \pi^v(\cdot|\tilde{s})\right],$$

$$J^i(\lambda, \chi_*) = \mathbb{E}\left[r^i(s, a, b) + \sum_{s'} p(s'|s, a, b) v^i(s') | b \sim \rho^v(\cdot|s), a \sim \pi_*^v(\cdot|\tilde{s})\right],$$

where $\pi^v, \rho^v$ are corresponding policies implied from behavioral strategies $\chi, \lambda$, respectively. Recall the definition of the minimax operator of $L^i v^i(s)$, we have, for all $s \in S$,

$$L^i v^i(s) = r^i_{(\pi_*^v, \rho_*^v)}(s) + \gamma \sum_{s' \in S} p_{(\pi_*^v, \rho_*^v)}(s'|s) v^i(s')$$

$\square$

Based on the proof, we also denote $(\pi_*^v, \rho_*^v)$ as an NE policy for the EFG $(v^1, \cdots, v^N, -v^1, \cdots, -v^N)$ for convenience, instead of calling it the joint policy derived from an NE for the EFG $(v^1, \cdots, v^N, -v^1, \cdots, -v^N)$.

We can get a corollary from Lemma A.7 that when optimal value functions of an MG-SPA exist, we are able to get a joint policy that satisfies the Bellman equations of the MG-SPA by computing an NE for a constructed EFG $(v_*^1, \cdots, v_*^N, -v_*^1, \cdots, -v_*^N)$. Later in Theorem 4.7, we show that a joint policy that satisfies the Bellman equations of an MG-SPA is in a robust equilibrium under Assumption 4.4. Therefore, Lemma A.7 provides insights into solving an MG-SPA by solving a corresponding EFG.

In the following proof, we aim to prove the existence of optimal value functions for an MG-SPA under Assumption 4.4.

## A.2  Proof of two propositions

**Proposition A.8** (Contraction mapping, same as proposition 4.5 in the main text.)**.** *Suppose $0 \leq \gamma < 1$ and Assumption 4.4 hold. Then $L$ is a contraction mapping on $\mathbb{V}$.*

*Proof.* Let $u$ and $v$ be in $\mathbb{V}$. Given Assumption 4.4, these two EFGs $(u^1, \cdots, u^N, -u^1, \cdots, -u^N)$, $(v^i, \cdots, v^N, -v^i, \cdots, -v^N)$ both have at least one mixed Nash equilibrium according to Lemma A.6. And let $(\pi_*^u, \rho_*^u)$ and $(\pi_*^v, \rho_*^v)$ be two Nash equilibriums for these two games, respectively. According to Lemma A.7, we have the following equations hold for all $s \in S$,

$$L^i v^i(s) = r^i_{(\pi_*^v, \rho_*^v)}(s) + \gamma \sum_{s' \in S} p_{(\pi_*^v, \rho_*^v)}(s'|s) v^i(s')$$

$$L^i u^i(s) = r^i_{(\pi_*^u, \rho_*^u)}(s) + \gamma \sum_{s' \in S} p_{(\pi_*^u, \rho_*^u)}(s'|s) u^i(s')$$

Then we have

$$r^i_{(\pi_*^u, \rho_*^v)}(s) + \gamma \sum_{s' \in S} p_{(\pi_*^u, \rho_*^v)}(s'|s) v^i(s') \leq L^i v^i(s) \leq r^i_{(\pi_*^v, \rho_*^u)}(s) + \gamma \sum_{s' \in S} p_{(\pi_*^v, \rho_*^u)}(s'|s) v^i(s'),$$

$$r^i_{(\pi_*^v, \rho_*^u)}(s) + \gamma \sum_{s' \in S} p_{(\pi_*^v, \rho_*^u)}(s'|s) u^i(s') \leq L^i u^i(s) \leq r^i_{(\pi_*^u, \rho_*^v)}(s) + \gamma \sum_{s' \in S} p_{(\pi_*^u, \rho_*^v)}(s'|s) u^i(s'),$$

since $(\pi_*^u, \rho_*^v)$ and $(\pi_*^v, \rho_*^u)$ are derived from the Nash equilibrium of the EFG $(v^i, \cdots, v^N, -v^i, \cdots, -v^N)$, and $(\pi_*^u, \rho_*^v)$ and $(\pi_*^v, \rho_*^u)$ are also derived from the Nash equilibrium of the EFG $(u^i, \cdots, u^N, -u^i, \cdots, -u^N)$. We assume that $L^i v^i(s) \leq L^i u^i(s)$, then we have

$$0 \leq L^i u^i(s) - L^i v^i(s)$$

$$\leq \left[ r^i_{(\pi_*^u, \rho_*^v)}(s) + \gamma \sum_{s' \in S} p_{(\pi_*^u, \rho_*^v)}(s'|s) u^i(s') \right] - \left[ r^i_{(\pi_*^u, \rho_*^v)}(s) + \gamma \sum_{s' \in S} p_{(\pi_*^u, \rho_*^v)}(s'|s) v^i(s') \right]$$

$$\leq \gamma \sum_{s' \in S} p_{(\pi_*^u, \rho_*^v)}(s'|s)(u^i(s') - v^i(s'))$$

$$\leq \gamma ||v^i - u^i||.$$

Repeating this argument in the case that $L^i u^i(s) \leq L^i v^i(s)$ implies that

$$||L^i v^i(s) - L^i u^i(s)|| \leq \gamma ||v^i - u^i||$$

for all $s \in S$, i.e. $L^i$ is a contraction mapping on $V$. Recall that $||v|| = \sup_j ||v^j||$, then we have

$$||Lv - Lu|| = \sup_j ||L^j v^j - L^j u^j|| \leq \gamma \sup_j ||v^j - u^j|| = \gamma ||v - u||.$$

$L$ is a contraction mapping on $\mathbb{V}$.

$\square$

**Proposition A.9** (Complete Space, same as proposition 4.6 in the main text.)**.** $\mathbb{V}$ *is a complete normed linear space.*

*Proof.* Recall that $\mathbb{V}$ denote the set of bounded real-valued functions on $S \times \cdots \times S$, i.e. the cross product of $N$ state set with component-wise partial order and norm $||v|| := \sup_{s \in S} \sup_j |v^i(s)|$. Since $\mathbb{V}$ is closed under addition and scalar multiplication and is endowed with a norm, it is a normed linear space. Since every Cauchy sequence contains a limit point in $\mathbb{V}$, $\mathbb{V}$ is a complete space. $\square$

## A.3 Proof of Theorem 4.7

In this section, our goal is to prove Theorem 4.7. We first prove (1) the optimal value function of an MG-SPA satisfies the Bellman equation by applying the Squeeze theorem [Theorem 3.3.6, Sohrab (2003)] in A.3.1. Then we prove that a unique solution of the Bellman equation exists using fixed-point theorem (Smart, 1980) in A.3.2. Thereby, the existence of the optimal value function gets proved. By introducing (3), we characterize the relationship between the optimal value function and a robust equilibrium. The proof of (3) can be found in A.3.3. However, (3) does not imply the existence of an RE. To this end, in (4), we formally establish the existence of RE when the optimal value function exists. We formulate a $2N$-player Extensive-form game (EFG) (Osborne & Rubinstein, 1994; Von Neumann & Morgenstern, 2007) based on the optimal value function such that its Nash equilibrium (NE) is equivalent to an RE of the MG-SPA. The details are in A.3.4.

---

**Theorem A.10** (Same as theorem 4.7 in the main text).

*Suppose $0 \leq \gamma < 1$ and Assumption 4.4 holds.*

*(1) (Solution of Bellman equation) A value function $v_* \in \mathbb{V}$ is an optimal value function if for all $i \in \mathcal{N}$, the point-wise value function $v_*^i \in V$ satisfies the corresponding Bellman Equation (6), i.e. $v_* = Lv_*$.*

*(2) (Existence and uniqueness of optimal value function) There exists a unique $v_* \in \mathbb{V}$ satisfying $Lv_* = v_*$, i.e. for all $i \in \mathcal{N}$, $L^i v_*^i = v_*^i$.*

*(3) (Robust equilibrium and optimal value function) A joint policy $d_* = (\pi_*, \rho_*)$, where $\pi_* = (\pi_*^1, \cdots, \pi_*^N)$ and $\rho_* = (\rho_*^{\tilde{1}}, \cdots, \rho_*^{\tilde{N}})$, is a robust equilibrium if and only if $v^{d_*}$ is the optimal value function.*

*(4) (Existence of robust equilibrium) There exists a mixed RE for an MG-SPA.*

---

### A.3.1 (1) Solution of Bellman equation

*Proof.* First, we prove that if there exists a $v^i \in V$ such that $v^i \geq Lv^i$ then $v^i \geq v_*^i$. $v^i \geq Lv^i$ implies $v^i \geq \max\min[r^i + \gamma Pv^i] = r_d^i + \gamma P_d v^i$, where $d = (\pi_*^{v,-i}, \pi_*^{v,i}, \rho_*^{v,-\tilde{i}}, \rho_*^{v,\tilde{i}})$ is a Nash equilibrium for the EFG $v = (v^1, \cdots, v^N, -v^1, \cdots, -v^N)$. We omit the superscript $v$ for convenience when there is no confusion. We choose a list of policy i.e. $z = (d_1, d_2, \cdots)$ where $d_j = (\pi_*^{-i}, \pi_j^i, \rho_*^{-\tilde{i}}, \rho^{\tilde{i}}_*)$. Then we have

$$v^i \geq r_{d_1} + \gamma P_{d_1} v^i \geq r_{d_1}^i + \gamma P_{d_1}(r_{d_2}^i + \gamma P_{d_2} v^i) = r_{d_1}^i + \gamma P_{d_1} r_{d_2}^i + \gamma P_{d_1} P_{d_2} v^i$$

By induction, it follows that, for $n \geq 1$,

$$v^i \geq r_{d_1}^i + \gamma P_{d_1} r_{d_2}^i + \cdots + \gamma^{n-1} P_{d_1} \cdots P_{d_{n-1}} r_{d_n}^i + \gamma^n P_z^n v^i$$

$$v^i - v_z^i \geq \gamma^n P_z^n v^i - \sum_{t=n}^{\infty} \gamma^t P_z^t r_{d_{t+1}}^i \tag{12}$$

Since $||\gamma^n P_z^n v^i|| \leq \gamma^n ||v^i||$ and $\gamma \in [0,1)$, for $\epsilon > 0$, we can find a sufficiently large $n$ such that

$$\epsilon e/2 \geq \gamma^n P_z^n v^i \geq -\epsilon e/2 \tag{13}$$

where $e$ denotes a vector of 1's. And as a result of Assumption 4.4-(1), we have

$$-\sum_{t=n}^{\infty} \gamma^t P_z^t r_{d_{t+1}}^i \geq -\frac{\gamma^n Me}{1-\gamma} \tag{14}$$

Then we have

$$v^i(s) - v_z^i(s) \geq -\epsilon \tag{15}$$

for all $s \in S$ and $\epsilon > 0$. Let all $d_j$ the same, since $\epsilon$ was arbitrary, we have

$$v^i(s) \geq \max_{\pi^i} \min_{\rho^i} v_z^i(s) = v_*^i(s) \tag{16}$$

Then we prove that if there exists a $v^i \in V$ such that $v^i \leq Lv^i$ then $v^i \leq v_*^i$. For arbitrary $\epsilon > 0$ there exists a joint policy $d' = (\pi_*^{-i}, \pi_*^i, \rho_*^{-i}, \rho^{\tilde{i}})$ and a list of policy $z = (d', d', \cdots)$ such that

$$v^i \leq r_{d'}^i + \gamma P_{d'} v^i + \epsilon$$
$$(I - \gamma P_{d'})v^i \leq r_{d'}^i + \epsilon$$
$$\leq (I - \gamma P_{d'})^{-1} r_{d'}^i + (1-\gamma)^{-1} \epsilon e = v_z^i + (1-\gamma)^{-1} \epsilon e$$
$$\leq v_*^i + (1-\gamma)^{-1} \epsilon e$$

The equality holds because the Theorem 6.1.1 in Puterman (2014). Since $\epsilon$ was arbitrary, we have

$$v^i \leq v_*^i \tag{17}$$

So if there exists a $v^i \in V$ such that $v^i = L^i v^i$ i.e. $v^i \leq L^i v^i$ and $v^i \geq L^i v^i$, we have $v^i = v_*^i$, i.e. if $v^i$ satisfies the Bellman equation, $v^i$ is an optimal value function.

$\square$

### A.3.2 (2) Existence of optimal value function

*Proof.* Proposition 4.5 and 4.6 establish that $\mathbb{V}$ is a complete normed linear space and $L$ is a contraction mapping, so that the hypothesis of Banach Fixed-Point Theorem are satisfied (Smart, 1980). Therefore there exists a unique solution $v_* \in \mathbb{V}$ to $Lv = v$. From (1), we know if $v_*$ satisfies the Bellman equation, it is an optimal value function. Therefore, the existence of the optimal value function is proved. $\square$

### A.3.3 (3) robust equilibrium and optimal value function

*Proof.* (i) robust equilibrium $\rightarrow$ Optimal value function.

Suppose $d^*$ is a robust equilibrium. Then $v^{d^*} = v^*$. From (2), it follows that $v^{d^*}$ satisfies $Lv = v$. Thus $v^{d^*}$ is the optimal value function.

(ii) Optimal value function $\rightarrow$ robust equilibrium.

Suppose $v^{d^*}$ is the optimal value function, i.e., $Lv^{d^*} = v^{d^*}$. The proof of (1) implies that $v^{d^*} = v^*$, so $d^*$ is in robust equilibrium. $\square$

### A.3.4 (4) Existence of robust equilibrium

*Proof.* From (2), we know that there exists a solution $v_* \in \mathbb{V}$ to the Bellman equation $Lv = v$. Now, we consider an EFG based on $(v_*^1, \cdots, v_*^N, -v_*^1, \cdots, -v_*^N)$. Under Assumption 4.4, we can get an NE policy $(\pi_*^{v_*}, \rho_*^{v_*})$ by solving the EFG as a consequence of Lemma A.6. According to Lemma A.7, $(\pi_*^{v_*}, \rho_*^{v_*})$ satisfies

$$L^i v_*^i(s) = r_{(\pi_*^{v_*}, \rho_*^{v_*})}^i(s) + \gamma \sum_{s' \in S} p_{(\pi_*^{v_*}, \rho_*^{v_*})}(s'|s) v_*^i(s'),$$

for all $s \in S$. According to (3), $(\pi_*^{v_*}, \rho_*^{v_*})$ is a robust equilibrium. $\square$

## A.4 Proof of Corollary 4.9.1

> **Corollary A.10.1** (Same as Corollary 4.9.1).
>
> *Theorem 4.7 still holds when all agents and adversaries in an MG-SPA use history-dependent policies with a finite time horizon.*

*Proof.* From subsection 4.3, we can find the main difference between history-dependent-policy-based RE and Markov-policy-based RE are the definitions and notations of policies and states. To prove Theorem 4.7, we construct an EFG based on the current state $s_t$. Similarly, to prove Corollary 4.9.1, we construct an EFG based on the concatenated states $s_{h,t}$ and $\tilde{s}_{h-1,t-1}$ that includes the current state $s_t$ and historical state information $s_{t-1}, \cdots, s_{t-h+1}, \tilde{s}_{t-1}, \cdots, \tilde{s}_{t-h+1}$. Notice that $h$ is a finite number. Hence, the concatenated state space is still finite. We now construct another extensive-form game in which a centralized nature player $(P1)$ has $|\tilde{S}|$ alternatives (branches) to choose from, whereas a centralized agent $(P2)$ has $|A|$ alternatives, and the order of play is that the centralized nature player acts before the centralized agent does. The set $A$ is the same as the agents' joint action set in an MG-SPA, set $\tilde{S}$ is a set of perturbed states constrained by a constrained parameter $\epsilon$.

> **Definition A.11.** *An extensive-form game based on $(v^1, \cdots, v^N, -v^1, \cdots, -v^N)$ under concatenated states $s_{h,t} = (s_t, \cdots, s_{t-h+1}) \in S_h$ and $\tilde{s}_{h,t-1} = (\tilde{s}_{t-1}, \cdots, \tilde{s}_{t-h+1}) \in S_{h-1}$ is a finite tree structure with:*
>
> 1. *A player $P1$ has a action set $\tilde{S} = \overbrace{\mathcal{B}(\epsilon, s) \times \cdots \times \mathcal{B}(\epsilon, s)}^{N}$, with a typical element designed as $\tilde{s}$. And $P1$ moves first.*
>
> 2. *Another player $P2$ has an action set $A$, with a typical element designed as $a$. And $P2$ which moves after $P1$.*
>
> 3. *A specific vertex indicating the starting point of the game.*
>
> 4. *A payoff function $g_s(\tilde{s}, a) = (g_s^1(\tilde{s}, a), \cdots, g_s^N(\tilde{s}, a))$ where $s = s_t = s_{h,t}[1]$ is the first element of $s_{h,t}$, $\tilde{s} = \tilde{s}_t \in \tilde{S}$, $g_s^i(\tilde{s}, a) = r^i(s, a, f_s^{-1}(\tilde{s})) + \sum_{s'} p(s'|a, f_s^{-1}(\tilde{s})) v^i(s')$ assigns a real number to each terminal vector of the tree. Player $P1$ gets $-g_s(\tilde{s}, a)$ while player $P2$ gets $g_s(\tilde{s}, a)$.*
>
> 5. *A partition of the nodes of the tree into two player sets (to be denoted by $\bar{N}^1$ and $\bar{N}^2$ for $P1$ and $P2$, respectively).*
>
> 6. *A sub-partition of each player set $\bar{N}^i$ into information set $\{\eta_j^i\}$, such that the same number of immediate branches emanates from every node belonging to the same information set, and no node follows another node in the same information set.*

The definitions of behavioral strategy and Nash equilibrium keep the same. Then the behavioral strategy $\lambda$ for $P1$ is a mapping from his information sets and action space into a probability simplex, i.e. $\lambda(\tilde{s}|s_{h,t} = (s_t, \cdots, s_{t-h+1}))$ is the probability of choosing $\tilde{s}$ given $s_{h,t}$. Similarly, the behavioral strategy $\chi$ for $P2$ is $\chi(a|\tilde{s}_{h,t} = (\tilde{s}, \tilde{s}_{t-1}, \cdots, \tilde{s}_{t-h+1}))$, i.e. the probability of choosing action $a$ when $\tilde{s}_{h,t}$ is given.

Now let us check the correctness of Lemma A.6 when EFG is constructed following Definition A.11. We can find that $S_h$ is a finite state space for any finite time horizon $h$ since $S_h$ is a product topology on finite spaces. Then Lemma A.6 still holds because the EFG degenerates to a zero-sum two-person extensive-form game with finite strategies and perfect recall.

Then let us check Lemma A.7. We re-write it in Lemma A.12 in which the behavioral strategy $\chi(a|\tilde{s}_{h,t})$ and $\lambda(\tilde{s}|s_{h,t})$ are used. The proof of Lemma A.12 is similar to that of Lemma A.7.

**Lemma A.12.** *Suppose $f$ is a bijection when $s$ is fixed and an NE $(\lambda_*, \chi_*)$ exists for an EFG $(v^1, \cdots, v^N, -v^1, \cdots, -v^N)$. We define a joint policy $(\pi_*^v, \rho_*^v)$ as the joint policy implied from the NE $(\lambda_*, \chi_*)$, where $\rho_*^v(b|s_h) = \lambda_*(\tilde{s} = f_s(b)|s_h), \pi_*^v(a|\tilde{s}_h = (f_s(b), \tilde{s}_{t-1}, \cdots, \tilde{s}_{t-h+1})) = \chi_*(a|\tilde{s}_h)$. Then the joint policy $(\pi_*^v, \rho_*^v)$ satisfies $L^i v^i(s) = r^i_{(\pi_*^v, \rho_*^v)}(s) + \gamma \sum_{s' \in S} p_{(\pi_*^v, \rho_*^v)}(s'|s) v^i(s')$ for all $s \in S$.*

*Proof.* The NE of the extensive-form game $(\lambda_*, \chi_*)$ implies that for all $i = 1, \cdots, N, \ s \in S, \lambda \in \Gamma^1, \chi \in \Gamma^2$, we have

$$J^i(\lambda, \chi_*) \geq J^i(\lambda_*, \chi_*) \geq J^i(\lambda_*, \chi),$$

where $J^i(\lambda, \chi) = \mathbb{E}[r^i(s, a, f_s^{-1}(\tilde{s})) + \sum_{s'} p(s'|s, a, f_s^{-1}(\tilde{s})) v^i(s') | \tilde{s} \sim \lambda(\cdot|s_h), a \sim \chi(\cdot|\tilde{s}_h)]$ according to Definition A.5. Let $b$ denote $f_s^{-1}(\tilde{s})$, because $f$ is a bijection when $s$ is fixed, $f_s(b) = (f_s(b^1), \cdots, f_s(b^N))$ is a bijection, and the inverse function $f_s^{-1}(\tilde{s}) = (f_s^{-1}(\tilde{s}^1), \cdots, f_s^{-1}(\tilde{s}^N))$ exists and is a bijection as well, then we have

$$
\begin{aligned}
J^i(\lambda_*, \chi_*) =& \mathbb{E}\left[r^i(s, a, f_s^{-1}(\tilde{s})) + \sum_{s'} p(s'|s, a, f_s^{-1}(\tilde{s})) v^i(s') | \tilde{s} \sim \lambda_*(\cdot|s_h), a \sim \chi_*(\cdot|\tilde{s}_h)\right] \\
=& \mathbb{E}\left[r^i(s, a, b) + \sum_{s'} p(s'|s, a, b) v^i(s') | b \sim \lambda_*(f_s(b)|s_h), a \sim \chi_*(\cdot|(f_s(b), \tilde{s}_{t-1}, \cdots, \tilde{s}_{t-h+1}))\right] \\
=& \mathbb{E}\left[r^i(s, a, b) + \sum_{s'} p(s'|s, a, b) v^i(s') | b \sim \rho_*^v(\cdot|s_h), a \sim \pi_*^v(\cdot|\tilde{s}_h)\right]
\end{aligned}
$$

Similarly, we have

$$J^i(\lambda_*, \chi) = \mathbb{E}\left[r^i(s, a, b) + \sum_{s'} p(s'|s, a, b) v^i(s') | b \sim \rho_*^v(\cdot|s_h), a \sim \pi^v(\cdot|\tilde{s}_h)\right],$$

$$J^i(\lambda, \chi_*) = \mathbb{E}\left[r^i(s, a, b) + \sum_{s'} p(s'|s, a, b) v^i(s') | b \sim \rho^v(\cdot|s_h), a \sim \pi_*^v(\cdot|\tilde{s}_h)\right],$$

where $\pi^v, \rho^v$ are corresponding policies implied from behavioral strategies $\chi, \lambda$, respectively. Recall the definition of the minimax operator of $L^i v^i(s)$, we have, for all $s \in S$,

$$L^i v^i(s) = r^i_{(\pi_*^v, \rho_*^v)}(s) + \gamma \sum_{s' \in S} p_{(\pi_*^v, \rho_*^v)}(s'|s) v^i(s')$$

$\square$

Proposition 4.6 still holds when agents and adversaries adopt history-dependent policies since we do not require Markov policies in the proof. Propositions 4.5 also holds which can be proved by utilizing the properties of NE for EFGs defined in Definition A.11. Specifically, in the proof, we use EFGs defined in Definition A.11 instead of Definition A.3. The subsequent proof of Propositions 4.5 keeps the same.

Then in the proof of Theorem 4.7, we are able to continue to utilize Propositions 4.6 and 4.5. Similarly, the EFGs used in the proof are replaced by Definition A.11. The definition of the minimax operator does not constrain the type of policies. The properties of the minimax operator can be continually used as well. The main body of proof keeps the same. Theorem 4.7 still holds when agents and adversaries adopt history-dependent policies.

$\square$

# B  Algorithm

## B.1  Robust multi-agent Q-learning (RMAQ)

In this section, we prove the convergence of RMAQ under certain conditions. First, let's recall the convergence theorem and certain assumptions.

---

**Assumption B.1** (Same as assumption 5.1).

*(1) State and action pairs have been visited infinitely often. (2) The learning rate $\alpha_t$ satisfies the following conditions: $0 \leq \alpha_t < 1$, $\sum_{t \geq 0} \alpha_t^2 \leq \infty$; if $(s, a, b) \neq (s_t, a_t, b_t)$, $\alpha_t(s, a, b) = 0$. (3) An NE of the EFG based on $(q_t^1, \cdots, q_t^N, -q_t^1, \cdots, -q_t^N)$ exists at each iteration $t$.*

---

**Theorem B.2** (Same as theorem 5.2).

*Under Assumption B.1, the sequence $\{q_t\}$ obtained from (18) converges to $\{q_*\}$ with probability 1, which are the optimal action-value functions that satisfy Bellman equations (5) for all $i = 1, \cdots, N$.*

$$q_{t+1}^i(s_t, a_t, b_t) = (1 - \alpha_t) q_t^i(s_t, a_t, b_t) + \tag{18}$$

$$\alpha_t \left[ r_t^i + \gamma \sum_{a_{t+1} \in A} \sum_{b_{t+1} \in B} \pi_{*,t}^{q_t}(a_{t+1}|\tilde{s}_{t+1}) \rho_{*,t}^{q_t}(b_{t+1}|s_{t+1}) q_t^i(s_{t+1}, a_{t+1}, b_{t+1}) \right],$$

---

*Proof.* Define the operator $Tq_t = T(q_t^1, \cdots, q_t^N) = (T^1 q_t^1, \cdots, T^N q_t^N)$ where the operator $T^i$ is defined as below:

$$T^i q_t^i(s, a, b) = r_t^i + \gamma \sum_{a' \in A} \sum_{b' \in B} \pi_*^{q_t}(a'|\tilde{s}') \rho_*^{q_t}(b'|s') q_t^i(s', a', b') \tag{19}$$

for $i \in \mathcal{N}$, where $(\pi_*^{q_t}, \rho_*^{q_t})$ is the tuple of Nash equilibrium policies for the EFG based on $(q_t^1, \cdots, q_t^N, -q_t^1, \cdots, -q_t^N)$ obtained from (18). Because of proposition B.3 and proposition B.4 the Lemma 8 in Hu & Wellman (2003) or Corollary 5 in Szepesvári & Littman (1999) tell that $q_{t+1} = (1 - \alpha_t) q_t + \alpha_t T q_t$ converges to $q_*$ with probability 1. $\qquad\square$

---

**Proposition B.3** (Contraction mapping).

$Tq_t = (T^1 q_t^1, \cdots, T^N q_t^N)$ *is a contraction mapping.*

---

*Proof.* We omit the subscript $t$ when there is no confusion. Assume $T^i p^i \geq T^i q^i$, we have

$$0 \leq T^i p^i - T^i q^i$$

$$= \gamma \left\| \sum_{a' \in A} \sum_{b' \in B} \pi_*^p(a'|\tilde{s}') \rho_*^p(b'|s') p^i(s', a', b') - \sum_{a' \in A} \sum_{b' \in B} \pi_*^q(a'|\tilde{s}') \rho_*^q(b'|s') q^i(s', a', b') \right\|$$

$$\leq \gamma \left\| \sum_{a' \in A} \sum_{b' \in B} \pi_*^q(a'|\tilde{s}') \rho_*^q(b'|s') p^i(s', a', b') - \sum_{a' \in A} \sum_{b' \in B} \pi_*^p(a'|\tilde{s}') \rho_*^p(b'|s') q^i(s', a', b') \right\|$$

$$\leq \gamma \left\| p^i - q^i \right\|. \tag{20}$$

Repeating the case $T^i p^i \leq T^i q^i$ implies that $T^i$ is a contraction mapping such that $\|T^i p^i - T^i q^i\| \leq \gamma \|p^i - q^i\|$ for all $p^i, q^i \in Q$. Recall that $\|p - q\| = \sup_j \|p^j - q^j\|$

$$\|Tp - Tq\| = \sup_j \|T^j p^j - T^j q^j\| \leq \gamma \sup_j \|p^j - q^j\| = \gamma \|p - q\|$$

$T$ is a contraction mapping such that $\|Tp - Tq\| \leq \gamma \|p - q\|$ for all $p, q \in \mathbb{Q}$. $\qquad\square$

**Proposition B.4** (A condition of Lemma 8 in Hu & Wellman (2003) also Corollary 5 in Szepesvári & Littman (1999))**.**

$$q_* = \mathbb{E}[Tq_*] \tag{21}$$

*Proof.*

$$
\begin{aligned}
\mathbb{E}\left[T^i q_*^i(s,a,b)\right] &= \mathbb{E}\left[r^i(s,a,b) + \gamma \sum_{a' \in A} \sum_{b' \in B} \pi_*(a'|\tilde{s}')\rho_*(b'|s')q_*^i(s',a',b')\right] \\
&= r^i(s,a,b) + \gamma \sum_{s' \in S} p(s'|s,a,b) \sum_{a' \in A} \sum_{b' \in B} \pi_*(a'|\tilde{s}')\rho_*(b'|s')q_*^i(s',a',b') \\
&= q_*^i(s,a,b) \tag{22}
\end{aligned}
$$

Therefore $q_* = \mathbb{E}[Tq_*]$. $\qquad\square$

## B.2 Robust multi-agent actor-critic (RMAAC)

In this section, we first give the details of policy gradients proof in MG-SPA and then list the Pseudo code of RMAAC.

### B.2.1 Proof of policy gradients

Recall the policy gradient in RMAAC for MG-SPA in the following:

---

**Theorem B.5** (Policy gradient in RMAAC for MG-SPA, same as the theorem 5.3)**.** *For each agent* $i \in \mathcal{N}$ *and adversary* $\tilde{i} \in \mathcal{M}$*, the policy gradients of the objective* $J^i(\theta, \omega)$ *with respect to the parameter* $\theta, \omega$ *are:*

$$\nabla_{\theta^i} J^i(\theta, \omega) = \mathbb{E}_{(s,a,b) \sim p(\pi, \rho)} \left[ q^{i, \pi, \rho}(s, a, b) \nabla_{\theta^i} \log \pi^i(a^i | \tilde{s}^i) \right] \tag{23}$$

$$\nabla_{\omega^i} J^i(\theta, \omega) = \mathbb{E}_{(s,a,b) \sim p(\pi, \rho)} \left[ q^{i, \pi, \rho}(s, a, b) [\nabla_{\omega^i} \log \rho^i(b^i | s) + reg] \right] \tag{24}$$

*where* $reg = \nabla_{\tilde{s}^i} \log \pi^i(a^i | \tilde{s}^i) \nabla_{b^i} f(s, b^i) \nabla_{\omega^i} \rho(b^i | s)$*.*

---

*Proof.* We first start with the derivative of the state value function on $\theta^i$:

$$\nabla_{\theta^i} v^{i, \pi, \rho}(s)$$

$$= \nabla_{\theta^i} \left[ \sum_{a \in A} \sum_{b \in B} \pi(a | \tilde{s}) \rho(b | s) q^{i, \pi, \rho}(s, a, b) \right]$$

$$= \sum_{a \in A} \sum_{b \in B} \left[ \nabla_{\theta^i} \pi(a | \tilde{s}) \rho(b | s) q^{i, \pi, \rho}(s, a, b) + \pi(a | \tilde{s}) \rho(b | s) \nabla_{\theta^i} q^{i, \pi, \rho}(s, a, b) \right]$$

$$= \sum_{a \in A} \sum_{b \in B} \left[ \nabla_{\theta^i} \pi(a | \tilde{s}) \rho(b | s) q^{i, \pi, \rho}(s, a, b) + \pi(a | \tilde{s}) \rho(b | s) \nabla_{\theta^i} \sum_{s', r} p(s', r | s, a, b)(r^i + v^{i, \pi, \rho}(s')) \right]$$

$$= \sum_{a \in A} \sum_{b \in B} \left[ \nabla_{\theta^i} \pi(a | \tilde{s}) \rho(b | s) q^{i, \pi, \rho}(s, a, b) + \pi(a | \tilde{s}) \rho(b | s) \nabla_{\theta^i} \sum_{s'} p(s' | s, a, b) v^{i, \pi, \rho}(s') \right] \tag{25}$$

We use $\phi^{\theta^i}(s)$ to denote $\sum_{a \in A} \sum_{b \in B} [\nabla_{\theta^i} \pi(a | \tilde{s}) \rho(b | s) q^{i, \pi, \rho}(s, a, b)]$. We use $p^{\pi, \rho}(s \to x, k)$ to denote the probability of transition from state $s$ to state $x$ with agents' joint policy $\pi$ and adversaries' joint policy $\rho$ after $k$ steps. For example, $p^{\pi, \rho}(s \to s, k = 0) = 1$ and $p^{\pi, \rho}(s \to s', k = 1) = \sum_{a \in A} \sum_{b \in B} \pi(a | \tilde{s}) \rho(b | s) p(s' | s, a, b)$. In the following proof, we sometimes use the superscript $i$ instead of $\tilde{i}$ to denote adversary $\tilde{i}$ when there is no confusion. Then we have:

$$\nabla_{\theta^i} v^{i, \pi, \rho}(s)$$

$$= \phi^{\theta^i}(s) + \sum_{a \in A} \sum_{b \in B} \pi(a | \tilde{s}) \rho(b | s) \nabla_{\theta^i} \sum_{s' \in S} p(s' | s, a, b) v^{i, \pi, \rho}(s')$$

$$= \phi^{\theta^i}(s) + \sum_{a \in A} \sum_{b \in B} \sum_{s' \in S} \pi(a | \tilde{s}) \rho(b | s) p(s' | s, a, b) \nabla_{\theta^i} v^{i, \pi, \rho}(s')$$

$$= \phi^{\theta^i}(s) + \sum_{s' \in S} p^{\pi, \rho}(s \to s', 1) \nabla_{\theta^i} v^{i, \pi, \rho}(s')$$

$$= \phi^{\theta^i}(s) + \sum_{s' \in S} p^{\pi, \rho}(s \to s', 1) \left[ \phi^{\theta^i}(s') + \sum_{s'' \in S} p^{\pi, \rho}(s' \to s'', 1) \nabla_{\theta^i} v^{i, \pi, \rho}(s'') \right]$$

$$= \cdots$$

$$= \sum_{x \in S} \sum_{k=0}^{\infty} p^{\pi, \rho}(s \to x, k) \phi^{\theta^i}(x) \tag{26}$$

By plugging in $\nabla_{\theta^i} v^{i,\pi,\rho}(s) = \sum_{x \in S} \sum_{k=0}^{\infty} p^{\pi,\rho}(s \to x, k)\phi^{\theta^i}(x)$ into the objective function $J^i(\theta, \omega)$, we can get the following results:

$$
\nabla_{\theta^i} J^i(\theta, \omega) = \nabla_{\theta^i} v^{i,\pi,\rho}(s_1)
$$

$$
= \sum_{s \in S} \sum_{k=0}^{\infty} p^{\pi,\rho}(s_1 \to s, k)\phi^{\theta^i}(s)
$$

$$
= \sum_{s \in S} \eta(s)\phi^{\theta^i}(s) \qquad \qquad ;Let\ \eta(s) = \sum_{k=0}^{\infty} p^{\pi,\rho}(s_1 \to s, k)\phi^{\theta^i}(s)
$$

$$
= \left(\sum_{s \in S} \eta(s)\right) \sum_{s \in S} \frac{\eta(s)}{\sum_{s \in S} \eta(s)}\phi^{\theta^i}(s)
$$

$$
\propto \sum_{s \in S} \frac{\eta(s)}{\sum_{s \in S} \eta(s)}\phi^{\theta^i}(s) \qquad \qquad ;\left(\sum_{s \in S} \eta(s)\right)\ is\ a\ constant
$$

$$
= \sum_{s \in S} \sum_{a \in A} \sum_{b \in B} d^{\pi,\rho}(s)\nabla_{\theta^i}\pi(a|\tilde{s})\rho(b|s)q^{i,\pi,\rho}(s, a, b) \qquad ;Let\ d^{\pi,\rho}(s) = \frac{\eta(s)}{\sum_{s \in S} \eta(s)}
$$

$$
= \sum_{s \in S} \sum_{a \in A} \sum_{b \in B} d^{\pi,\rho}(s)\frac{\nabla_{\theta^i}\pi(a|\tilde{s})}{\pi(a|\tilde{s})}\rho(b|s)q^{i,\pi,\rho}(s, a, b)\pi(a|\tilde{s})
$$

$$
= \mathbb{E}_{(s,a,b) \sim p(\pi,\rho)}\left[q^{i,\pi,\rho}(s, a, b)\nabla_{\theta^i}\log\pi^i(a^i|\tilde{s}^i)\right] \tag{27}
$$

Now we calculate the derivative of the state value function on $\omega^i$:

$$
\nabla_{\omega^i} v^{i,\pi,\rho}(s)
$$

$$
= \nabla_{\omega^i}\left[\sum_{a \in A} \sum_{b \in B} \pi(a|\tilde{s})\rho(b|s)q^{i,\pi,\rho}(s, a, b)\right]
$$

$$
= \sum_{a \in A} \sum_{b \in B}\left[\nabla_{\omega^i}\rho(b|s)\pi(a|\tilde{s})q^{i,\pi,\rho}(s, a, b) + \nabla_{\omega^i}\pi(a|\tilde{s})\rho(b|s)q^{i,\pi,\rho}(s, a, b) + \rho(b|s)\pi(a|\tilde{s})\nabla_{\omega^i}q^{i,\pi,\rho}(s, a, b)\right] \tag{28}
$$

We let $\psi^{\omega^i}(s) = \sum_{a \in A} \sum_{b \in B}\left[\nabla_{\omega^i}\rho(b|s)\pi(a|\tilde{s})q^{i,\pi,\rho}(s, a, b)\right]$ and $\phi^{\omega^i}(s) = \sum_{a \in A} \sum_{b \in B}\left[\nabla_{\omega^i}\pi(a|\tilde{s})\rho(b|s)q^{i,\pi,\rho}(s, a, b)\right]$. Similar to $\nabla_{\theta^i} v^{i,\pi,\rho}(s)$, we have:

$$
\nabla_{\omega^i} v^{i,\pi,\rho}(s)
$$

$$
= \psi^{\omega^i}(s) + \phi^{\omega^i}(s) + \sum_{a \in A} \sum_{b \in B}\left[\pi(a|\tilde{s})\rho(b|s)\nabla_{\theta^i}\sum_{s' \in S} p(s'|s, a, b)v^{i,\pi,\rho}(s')\right]
$$

$$
= \psi^{\omega^i}(s) + \phi^{\omega^i}(s) + \sum_{a \in A} \sum_{b \in B} \sum_{s' \in S} \pi(a|\tilde{s})\rho(b|s)p(s'|s, a, b)\nabla_{\omega^i} v^{i,\pi,\rho}(s')
$$

$$
= \psi^{\omega^i}(s) + \phi^{\omega^i}(s) + \sum_{s' \in S} p^{\pi,\rho}(s \to s', 1)\nabla_{\omega^i} v^{i,\pi,\rho}(s')
$$

$$
= \psi^{\omega^i}(s) + \phi^{\omega^i}(s) + \sum_{s' \in S} p^{\pi,\rho}(s \to s', 1)\left[\psi^{\omega^i}(s') + \phi^{\omega^i}(s') + \sum_{s' \in S} p^{\pi,\rho}(s' \to s'', 1)\nabla_{\omega^i} v^{i,\pi,\rho}(s'')\right]
$$

$$
= \cdots
$$

$$
= \sum_{x \in S} \sum_{k=0}^{\infty} p^{\pi,\rho}(s \to x, k)[\psi^{\omega^i}(s) + \phi^{\omega^i}(s)] \tag{29}
$$

By plugging in $\nabla_{\omega^i} v^{i,\pi,\rho}(s) = \sum_{x \in S} \sum_{k=0}^{\infty} p^{\pi,\rho}(s \to x, k)[\psi^{\omega^i}(s) + \phi^{\omega^i}(s)]$ into the objective function $J^i(\theta, \omega)$, we can get the following results:

$$
\begin{aligned}
\nabla_{\omega^i} J^i(\theta, \omega) &= \nabla_{\omega^i} v^{i,\pi,\rho}(s_1) \\
&= \sum_{s \in S} \sum_{k=0}^{\infty} p^{\pi,\rho}(s_1 \to s, k) \left[ \psi^{\omega^i}(s) + \phi^{\omega^i}(s) \right] \\
&\propto \sum_{s \in S} \sum_{a \in A} \sum_{b \in B} d^{\pi,\rho}(s) \left[ \nabla_{\omega^i} \pi(a|\tilde{s}) \rho(b|s) q^{i,\pi,\rho}(s, a, b) + \nabla_{\omega^i} \rho(b|s) \pi(a|\tilde{s}) q^{i,\pi,\rho}(s, a, b) \right] \\
&= \mathbb{E}_{(s,a,b) \sim p(\pi,\rho)} \left[ q^{i,\pi,\rho}(s, a, b) \nabla_{\omega^i} \log \rho(b|s) + q^{i,\pi,\rho}(s, a, b) \nabla_{\omega^i} \log \pi(a|\tilde{s}) \right] \\
&= \mathbb{E}_{(s,a,b) \sim p(\pi,\rho)} \left[ q^{i,\pi,\rho}(s, a, b) \nabla_{\omega^i} \log \rho^i(b^i|s) + q^{i,\pi,\rho}(s, a, b) \nabla_{\omega^i} \log \pi^i(a^i|\tilde{s}^i) \right] \\
&= \mathbb{E}_{(s,a,b) \sim p(\pi,\rho)} \left[ q^{i,\pi,\rho}(s, a, b) \nabla_{\omega^i} \log \rho^i(b^i|s) + q^{i,\pi,\rho}(s, a, b) \frac{\nabla_{\tilde{s}^i} \pi^i(a^i|\tilde{s}^i) \nabla_{b^i} f(s, b^i) \nabla_{\omega^i} \rho(b^i|s)}{\pi^i(a^i|\tilde{s}^i)} \right] \\
&= \mathbb{E}_{(s,a,b) \sim p(\pi,\rho)} \left\{ q^{i,\pi,\rho}(s, a, b)[\nabla_{\omega^i} \log \rho^i(b^i|s) + \nabla_{\tilde{s}^i} \log \pi^i(a^i|\tilde{s}^i) \nabla_{b^i} f(s, b^i) \nabla_{\omega^i} \rho(b^i|s)] \right\}
\end{aligned}
\tag{30}
$$

$\square$

### B.2.2 Policy gradients for deterministic polices

**Theorem B.6** (Policy gradients for deterministic polices in RMAAC for MG-SPA). *For each agent $i \in \mathcal{N}$ and adversary $\tilde{i} \in \mathcal{M}$ using deterministic policies, the policy gradients of the objective $J^i(\theta, \omega)$ with respect to the parameter $\theta, \omega$ are:*

$$
\nabla_{\theta^i} J^i(\theta, \omega) = \frac{1}{T} \sum_{t=1}^{T} \nabla_{a^i} q^i(s_t, a_t, b_t) \nabla_{\theta^i} \pi^i(\tilde{s}_t^i)|_{a_t^i = \pi^i(\tilde{s}_t^i), b_t^i = \rho^i(s_t)}
\tag{31}
$$

$$
\nabla_{\omega^i} J^i(\theta, \omega) = \frac{1}{T} \sum_{t=1}^{T} \left[ \nabla_{b^i} q^i(s_t, a_t, b_t) + reg \right] \nabla_{\omega^i} \rho^i(s_t)|_{a_t^i = \pi^i(\tilde{s}_t^i), b_t^i = \rho^i(s_t)}
\tag{32}
$$

*where $reg = \nabla_{b_t^i} f(s_t, b_t^i) \nabla_{a^i} q^i(s_t, a_t, b_t) \nabla_f \pi^i(f)$.*

*Proof.* Note that we here parameterize all policies $\pi^i, \rho^{\tilde{i}}$ as deterministic policies. Then we have:

$$
\begin{aligned}
\nabla_{\theta^i} J^i(\theta, \omega) &= \mathbb{E}_{s \sim p_{(\pi,\rho)}} \left[ \nabla_{\theta^i} q^i(s, a, b) \right] \\
&= \mathbb{E}_{s \sim p_{(\pi,\rho)}} \left[ \nabla_{a^i} q^i(s, a, b) \nabla_{\theta^i} \pi^i(\tilde{s}^i) \right], \\
\nabla_{\omega^i} J^i(\theta, \omega) &= \mathbb{E}_{s \sim p_{(\pi,\rho)}} \left[ \nabla_{\omega^i} q^i(s, a, b) \right] \\
&= \mathbb{E}_{s \sim p_{(\pi,\rho)}} \left[ \nabla_{a^i} q^i(s, a, b) \nabla_{\tilde{s}^i} \pi^i(\tilde{s}^i) \nabla_{b^i} f(s^i, b^i) \nabla_{\omega^i} \rho^i(s) + \nabla_{b^i} q^i(s, a, b) \nabla_{\omega^i} \rho^i(s) \right] \\
&= \mathbb{E}_{s \sim p_{(\pi,\rho)}} \left[ \nabla_{\omega^i} \rho^i(s) \left[ \nabla_{b^i} q^i(s, a, b) + reg \right] \right],
\end{aligned}
\tag{33}
$$
$$
\tag{34}
$$

where $reg = \nabla_{a^i} q^i(s, a, b) \nabla_{\tilde{s}^i} \pi^i(\tilde{s}^i) \nabla_{b^i} f(s, b^i)$. When the actors are updated in a mini-batch fashion (Mnih et al., 2015; Li et al., 2014), (9) and (10) approximate (33) and (34), respectively. $\square$

### B.2.3 Pseudo code of RMAAC

We provide the Pseudo code of RMAAC with deterministic policies in Algorithm 1. The stochastic policy version RMAAC is similar to Algorithm 1 but uses different policy gradients.

---

**Algorithm 1:** RMAAC with deterministic policies

---

**1** Randomly initialize the critic network $q^i(s, a, b|\eta^i)$, the actor network $\pi^i(\cdot|\theta^i)$, and the adversary network $\rho^i(\cdot|\omega^i)$ for agent $i$. Initialize target networks $q^{i\prime}, \pi^{i\prime}, \rho^{i\prime}$;

**2** **for** *each episode* **do**

**3**      Initialize a random process $\mathcal{N}$ for action exploration;

**4**      Receive initial state $s$;

**5**      **for** *each time step* **do**

**6**          For each adversary $i$, select action $b^i = \rho^i(s) + \mathcal{N}$ w.r.t the current policy and exploration. Compute the perturbed state $\tilde{s}^i = f(s, b^i)$. Execute actions $a^i = \pi(\tilde{s}^i) + \mathcal{N}$ and observe the reward $r = (r^1, ..., r^n)$ and the new state information $s'$ and store$(s, a, b, \tilde{s}, r, s')$ in replay buffer $\mathcal{D}$. Set $s' \rightarrow s$;

**7**          **for** *agent i=1 to n* **do**

**8**              Sample a random minibatch of $K$ samples $(s_k, a_k, b_k, r_k, s'_k)$ from $\mathcal{D}$;

**9**              Set $y^i_k = r^i_k + \gamma q^{i\prime}(s'_k, a'_k, b'_k)|_{a^{i\prime}_k = \pi^{i\prime}(\tilde{s}^i_k), b^{i\prime}_k = \rho^{i\prime}(s_k)}$;

**10**              Update critic by minimizing the loss $\mathcal{L} = \frac{1}{K} \sum_k \left[ y^i_k - q^i(s_k, a_k, b_k) \right]^2$;

**11**              **for** *each iteration step* **do**

**12**                  Update actor $\pi^i(\cdot|\theta^i)$ and adversary $\rho^i(\cdot|\omega^i)$ using the following gradients

**13**                  $\theta^i \leftarrow \theta^i + \alpha_a \frac{1}{K} \sum_k \nabla_{\theta^i} \pi^i(\tilde{s}^i_k) \nabla_{a^i} q^i(s_k, a_k, b_k)$ where $a^i_k = \pi^i(\tilde{s}^i_k)$, $b^i_k = \rho^i(s_k)$;

**14**                  $\omega^i \leftarrow \omega^i - \alpha_b \frac{1}{K} \sum_k \nabla_{\omega^i} \rho^i(s_k) \left[ \nabla_{b^i} q^i(s_k, a_k, b_k) + reg \right]$ where $reg = \nabla_{a^i_k} q^i(s_k, a_k, b_k) \nabla_{\tilde{s}^i_k} \pi^i(\tilde{s}^i_k)$, $a^i_k = \pi^i(\tilde{s}^i_k)$, $b^i_k = \rho^i(s_k)$;

**15**              **end**

**16**          **end**

**17**          Update all target networks: $\theta^{i\prime} \leftarrow \tau\theta^i + (1 - \tau)\theta^{i\prime}$, $\omega^{i\prime} \leftarrow \tau\omega^i + (1 - \tau)\omega^{i\prime}$.

**18**      **end**

**19** **end**

---

## C  Experiments

### C.1  Robust multi-agent Q-learning (RMAQ)

In this section, we first introduce the designed two-player game in that the reward function and transition probability function are formally defined. The MG-SPA based on the two-player game is also further explained. Then we show more experimental results about the proposed robust multi-agent Q-learning (RMAQ) algorithm, such as the training process of the RMAQ algorithm in terms of the total discounted rewards.

### C.1.1  Two-player game

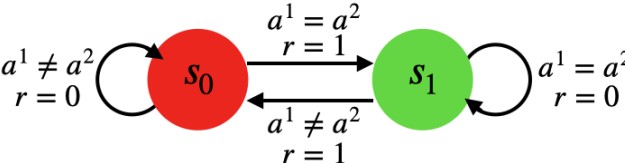

Figure 10: Two-player game: each player has two states and the same action set with size 2. Under state $s_0$, two players get the same reward 1 when they choose the same action. At state $s_1$, two players get the same reward 1 when they choose different actions. One state switches to another state only when two players get a reward, i.e. two players always stay in the current state until they get the reward.

Look at Figure 10 (same as Figure 3 in the main text.), this is how we run the designed two-player game. The reward function $r$ and transition probability function $p$ are defined as follows.

These two players get the same rewards all the time, i.e. they share a reward function $r$.

$$
r^i(s, a^1, a^2) = \begin{cases} 1, & a^1 = a^2, \text{and } s = s_0 \\ 1, & a^1 \neq a^2, \text{and } s = s_1 \\ 0, & a^1 \neq a^2, \text{and } s = s_0 \\ 0, & a^1 = a^2, \text{and } s = s_1 \end{cases} \tag{35}
$$

The state does not change until these two players get a positive reward. So the transition probability function $p$ is

$$
p(s_1|s, a^1, a^2) = \begin{cases} 1, & a^1 = a^2, \text{and } s = s_0 \\ 0, & a^1 \neq a^2, \text{and } s = s_0 \\ 1, & a^1 = a^2, \text{and } s = s_1 \\ 0, & a^1 \neq a^2, \text{and } s = s_1 \end{cases} \qquad p(s_0|s, a^1, a^2) = \begin{cases} 0, & a^1 = a^2, \text{and } s = s_0 \\ 1, & a^1 \neq a^2, \text{and } s = s_0 \\ 0, & a^1 = a^2, \text{and } s = s_1 \\ 1, & a^1 \neq a^2, \text{and } s = s_1 \end{cases} \tag{36}
$$

Possible Nash Equilibrium can be $\pi_1^* = (\pi_1^1, \pi_1^2)$ or $\pi_2^* = (\pi_2^1, \pi_2^2)$ where

$$
\pi_1^1(a^1|s) = \begin{cases} 1, & a^1 = 1, \text{and } s = s_0 \\ 0, & a^1 = 0, \text{and } s = s_0 \\ 1, & a^1 = 1, \text{and } s = s_1 \\ 0, & a^1 = 0, \text{and } s = s_1 \end{cases} \qquad \pi_1^2(a^2|s) = \begin{cases} 1, & a^2 = 1, \text{and } s = s_0 \\ 0, & a^2 = 0, \text{and } s = s_0 \\ 0, & a^2 = 1, \text{and } s = s_1 \\ 1, & a^2 = 0, \text{and } s = s_1 \end{cases} \tag{37}
$$

$$
\pi_2^1(a^1|s) = \begin{cases} 0, & a^1 = 1, \text{and } s = s_0 \\ 1, & a^1 = 0, \text{and } s = s_0 \\ 0, & a^1 = 1, \text{and } s = s_1 \\ 1, & a^1 = 0, \text{and } s = s_1 \end{cases} \qquad \pi_2^2(a^2|s) = \begin{cases} 0, & a^2 = 0, \text{and } s = s_0 \\ 1, & a^2 = 1, \text{and } s = s_0 \\ 1, & a^2 = 0, \text{and } s = s_1 \\ 0, & a^2 = 1, \text{and } s = s_1 \end{cases} \tag{38}
$$

NE $\pi_1^*$ means player 1 always selects action 1, player 2 selects action 1 under state $s_0$ and action 0 under state $s_1$. NE $\pi_2^*$ means player 1 always selects action 0, player 2 selects action 0 under state $s_0$ and action 0 under state $s_1$.

According to the definition of MG-SPA, we add two adversaries for each player to perturb the player's observations. And adversaries get negative rewards of players. We let adversaries share a same action space $B^1 = B^2 = \{0,1\}$, where 0 means do not disturb, 1 means change the observation to the opposite one. Therefore, the perturbed function $f$ in this MG-SPA is defined as:

$$
\begin{cases}
f(s_0, b = 0) = s_0 \\
f(s_1, b = 0) = s_1 \\
f(s_0, b = 1) = s_1 \\
f(s_1, b = 1) = s_0
\end{cases}
\tag{39}
$$

Obviously, $f$ is a bijective function when $s$ is given. And the constraint parameter $\epsilon = ||S||$, where $||S|| := \max |s - s'|_{\forall s, s' \in S}$, i.e. no constraints for adversaries' power.

A Robust Equilibrium (RE) of this MG-SPA would be $\tilde{d}^* = (\tilde{\pi}_*^1, \tilde{\pi}_*^2, \tilde{\rho}_*^1, \tilde{\rho}_*^2)$, where

$$
\begin{cases}
\tilde{\pi}_*^1(a^1|s) = 0.5, & \forall s \in S \\
\tilde{\pi}_*^2(a^2|s) = 0.5, & \forall s \in S \\
\tilde{\rho}_*^1(b^1|s) = 0.5, & \forall s \in S \\
\tilde{\rho}_*^2(b^2|s) = 0.5, & \forall s \in S
\end{cases}
\tag{40}
$$

### C.1.2 Training procedure

In Figure 11, we show the total discounted rewards in the function of training episodes. We set the learning rate as 0.1 and train our RMAQ algorithm for 400 episodes. And each episode contains 25 training steps. We can see the total discounted rewards converges to 50, i.e. the optimal value in the MG-SPA, after about 280 episodes or 7000 steps.

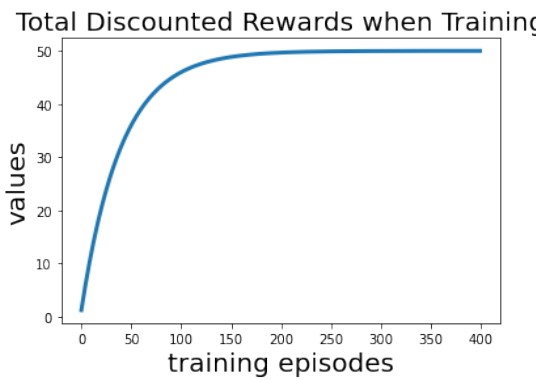

Figure 11: The total discounted rewards converge to the optimal value after about 280 training episodes.

## C.2 Robust multi-agent actor-critic (RMAAC)

In this section, we first briefly introduce the multi-agent environments we use in our experiments. Then we provide more experimental results and explanations, such as the testing results under a cleaned environment (accurate state information can be attained) and a randomly perturbed environment (injecting standard Gaussian noise in agents' observations). In the last subsection, we list all hyper-parameters we used in the experiments, as well as the baseline source code.

### C.2.1 Multi-agent environments

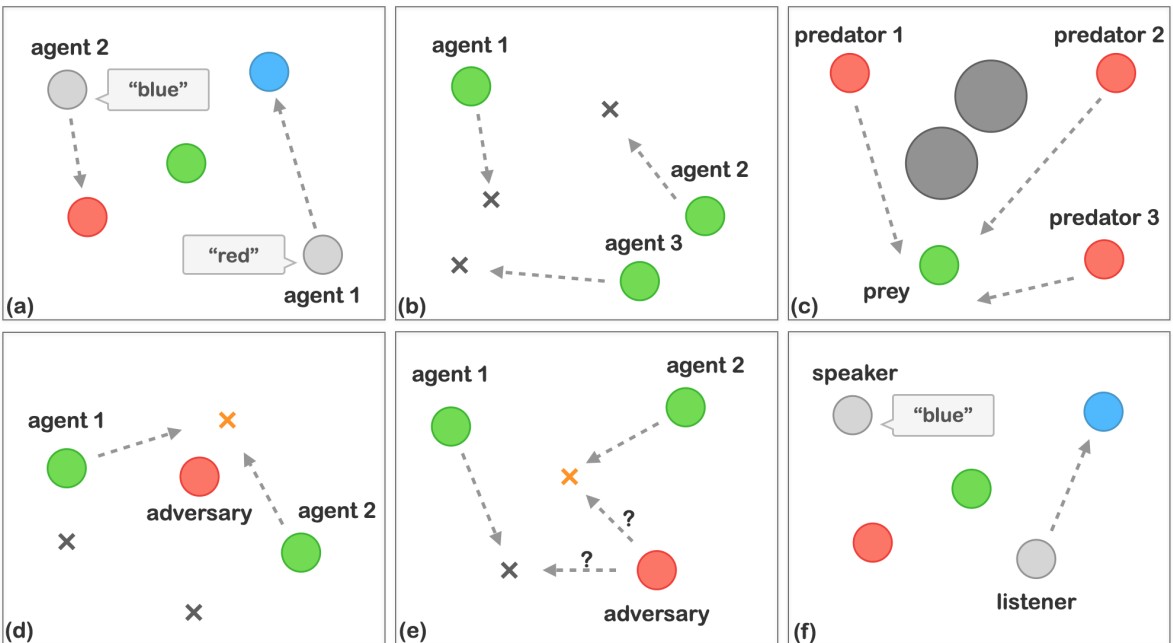

Figure 12: Illustrations of the experimental scenarios and some games we consider, including a) *Cooperative communication* b) *Cooperative navigation* c) *Predator prey* d) *Keep away* e) *Physical deception* f) *Navigate communication*

**Cooperative communication (CC):** This is a cooperative game. There are 2 agents and 3 landmarks of different colors. Each agent wants to get to their target landmark, which is known only by other agents. The reward is collective. So agents have to learn to communicate the goal of the other agent, and navigate to their landmark.

**Cooperative navigation (CN):** This is a cooperative game. There are 3 agents and 3 landmarks. Agents are rewarded based on how far any agent is from each landmark. Agents are penalized if they collide with other agents. So, agents have to learn to cover all the landmarks while avoiding collisions.

**Physical deception (PD):** This is a mixed cooperative and competitive task. There are 2 collaborative agents, 2 landmarks, and 1 adversary. Both the collaborative agents and the adversary want to reach the target, but only collaborative agents know the correct target. The collaborative agents should learn a policy to cover all landmarks so that the adversary does not know which one is the true target.

**Keep away (KA):** This is a competitive task. There is 1 agent, 1 adversary, and 1 landmark. The agent knows the position of the target landmark and wants to reach it. The adversary is rewarded if it is close to the landmark, and if the agent is far from the landmark. The adversary should learn to push the agent away from the landmark.

**Predator prey (PP):** This is a mixed game known as predator-prey. Prey agents (green) are faster and want to avoid being hit by adversaries (red). Predators are slower and want to hit good agents. Obstacles (large black circles) block the way.

**Navigate communication (NC):** This is a cooperative game that is similar to Cooperative communication. There are 2 agents and 3 landmarks of different colors. An agent is the 'speaker' that does not move but observes the goal of another agent. Another agent is the listener that cannot speak, but must navigate to the correct landmark.

**Predator prey+ (PP+):** This is an extension of the Predator prey environment by adding more agents. There are 2 preys, 6 adversaries, and 4 landmarks. Prey agents are faster and want to avoid being hit by adversaries. Predators are slower and want to hit good agents. Obstacles block the way.

### C.2.2   Experiments hyper-parameters

In Table 4, we show all hyper-parameters we use to train our policies and baselines. We also provide our source code in the supplementary material. The source code of M3DDPG (Li et al., 2019) and MADDPG (Lowe et al., 2017) accept the MIT License which allows any person obtaining them to deal in the code without restriction, including without limitation the rights to use, copy, modify, etc. More information about this license refers to `https://github.com/openai/maddpg` and `https://github.com/dadadidodi/m3ddpg`.

Table 4: Hyper-parameters

| Parameter | RMAAC | M3DDPG | MADDPG |
|---|---|---|---|
| optimizer | Adam | Adam | Adam |
| learning rate | 0.01 | 0.01 | 0.01 |
| adversarial learning rate | 0.005 | / | / |
| discount factor | 0.95 | 0.95 | 0.95 |
| replay buffer size | $10^6$ | $10^6$ | $10^6$ |
| number of hidden layers | 2 | 2 | 2 |
| activation function | Relu | Relu | Relu |
| number of hidden unites per layer | 64 | 64 | 64 |
| number of samples per minibatch | 1024 | 1024 | 1024 |
| target network update coefficient $\tau$ | 0.01 | 0.01 | 0.01 |
| iteration steps | 20 | 20 | 20 |
| constraint parameter $\epsilon$ | 0.5 | / | / |
| episodes in training | 10k | 10k | 10k |
| time steps in one episode | 25 | 25 | 25 |

### C.2.3   More testing results

In this subsection, we provide more testing results under a cleaned environment (accurate state information can be attained) and a randomly disturbed environment (injecting standard Gaussian noise into agents' observations).

As we have reported the comparison of mean episode testing rewards under a cleaned environment by using 4 different methods in the main manuscript (Figures 6 and 8), we further report the variance of testing results in the appendix. In Table 5 and 6, we also report the variances of testing rewards in different scenarios under different environment settings. Our method has the lowest variance in three of the five scenarios. Notice that RM1 denotes our RMAAC policy trained with the linear noise format $f_1$, RM2 denotes our RMAAC policy trained with the Gaussian noise format $f_2$, MA denotes MADDPG (`https://github.com/openai/maddpg`), M3 denotes M3DDPG (`https://github.com/dadadidodi/m3ddpg`).

MAPPO is a multi-agent reinforcement learning algorithm that performs well in cooperative multi-agent settings (Yu et al., 2021a). We use MP to denote MAPPO (`https://github.com/marlbenchmark/on-policy.`). In Figure 13, we compare its performance with our RMAAC algorithm in two cooperative scenarios of MPE. The details of scenarios such as Cooperative navigation, Navigate communication can be found in the last section. We can see that under the optimally perturbed environment, RMAAC outperforms MAPPO in all scenarios. Additionally, the reason we included MAPPO in the Appendix but not in the main text

is that the current source code provider of MAPPO only provides instructions and codes for using it in cooperative environments. However, to validate our proposed method in different settings, we carefully selected experimental environments to include different game types: cooperative, competitive, and mixed. Due to the lack of MAPPO source codes/implementation in competitive and mixed environments, if we were to include the experimental results of MAPPO in the main text, it could disrupt the integrity and uniformity of the experiment section in the main text. Therefore, we included them in the appendix as supplementary content.

In Figure 14 and Table 7, we compare the mean episode testing rewards and variances under different environments in the complicated scenario with a large number of agents between different algorithms. We adopt the Gaussian noise format in training RMAAC polices. We can see our method has the lowest variance under two of three environments and has the highest rewards under all environments.

Table 5: Variance of testing rewards under cleaned environment

| Algorithms | RM with $f_1$ | RM with $f_2$ | M3 | MA |
|---|---|---|---|---|
| Cooperative communication (CC) | 0.383 | 0.376 | **0.295** | 0.328 |
| Cooperative navigation (CN) | 0.413 | **0.361** | 0.416 | 0.376 |
| Physical deception (PD) | 0.175 | 0.165 | **0.133** | 0.143 |
| Keep away (KA) | 0.137 | **0.134** | 0.17 | 0.145 |
| Predator prey (PP) | 5.139 | **1.450** | 4.681 | 4.725 |

Table 6: Variance of testing rewards under randomly perturbed environment

| Algorithms | RM with $f_1$ | RM with $f_2$ | M3 | MA |
|---|---|---|---|---|
| Cooperative communication (CC) | 0.592 | **0.547** | 1.187 | 0.937 |
| Cooperative navigation (CN) | 0.336 | 0.33 | 0.328 | **0.321** |
| Physical deception (PD) | 0.222 | 0.292 | 0.209 | **0.184** |
| Keep away (KA) | **0.155** | **0.155** | 0.166 | 0.161 |
| Predator prey (PP) | 4.629 | **2.752** | 3.644 | 2.9 |

Table 7: Variance of testing rewards under different environments in Predator prey+.

| Algorithm | RM | M3 | MA |
|---|---|---|---|
| Optimally Perturbed Env | 4.199 | 4.046 | **3.924** |
| Randomly Perturbed Env | **4.664** | 5.774 | 6.191 |
| Cleaned Env | **3.928** | 5.521 | 6.006 |

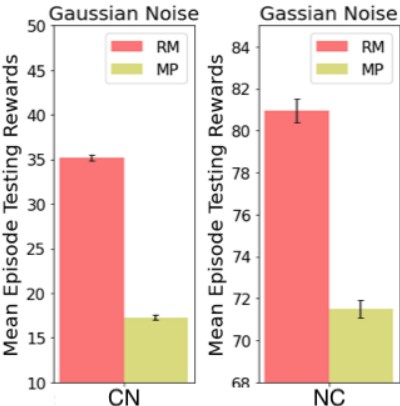

Figure 13: Comparison of mean episode testing rewards using MAPPO and RMAAC under optimally perturbed environments. RMAAC outperforms MAPPO in all cooperative scenarios under optimally perturbed environments.

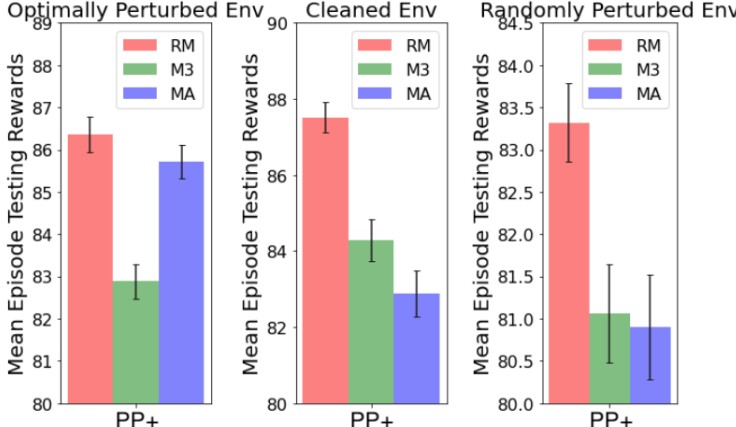

Figure 14: Comparison of mean episode testing rewards using different algorithms under different environments in Predator prey+. RMAAC outperforms all MARL baseline algorithms in the complicated multi-agent scenario.

## C.3 Ablation Study for RMAAC

In this subsection, we first investigate the effect of using different values of constraint parameters $\epsilon$ in the implementation of the RMAAC algorithm, then the effect of using different values of variance $\sigma$. Finally, we study the performance of the RMAAC algorithm under other types of attacks.

### C.3.1 Training Results Using Linear Noise with Different Constraint Parameters

**Training Setup:** In this subsection, we train several RMAAC policies using linear noise format as the state perturbation function, i.e. $f_1(s, b^{\tilde{i}}) = s + b^{\tilde{i}}$. The constraint parameter $\epsilon$ is respectively set as $0.01, 0.05, 0.1, 0.5, 1$ and $2$, given other hyper-parameters unchanged. Other used hyper-parameters can be found in Table 4.

**Training Results:** In Figure 15, 16, and 17, we show the training process in three scenarios: Cooperative communication (CC), Cooperative navigation (CN), Predator Prey (PP), respectively. The y-axis denotes the mean episode reward of the agents and the x-axis denotes the training episodes.

From these figures, we can see that, in general, the smaller the used variance, the higher the mean episode rewards RMAAC can achieve. However, RMAAC has different sensitivities to the value of variance in different scenarios. When we use $\epsilon = 2$, the RMAAC policies have the lowest mean episode rewards in all

three scenarios. Nevertheless, when we use the smallest constraint parameter $\epsilon = 0.01$, the trained RMAAC policies do not achieve the highest mean episode rewards in all three scenarios. In these three scenarios, it is clear to see the performance of RMAAC using $\epsilon = 0.5$ is better than or similar to the performance of RMAAC using $\epsilon = 1$, and better than the performance of RMAAC using $\epsilon = 2$, i.e. Performance($\epsilon = 0.5$) $\geq$ Performance($\epsilon = 1$) $>$ Performance($\epsilon = 2$). The performance of the RMAAC policies is close when the constraint parameters are less or equal to than 0.1.

### C.3.2 Testing Results Using Linear Noise with Different Constraint Parameters

In this subsection, we test well-trained RMAAC policies in perturbed environments where adversaries adopt linear noise format and different constraint parameters.

**Testing Setup:** The tested policy $\pi_{test}$ is trained with the linear noise format $f_1(s, b^{\tilde{i}}) = s + b^{\tilde{i}}$, constraint parameter is 0.5, where $b^{\tilde{i}} = \rho^{\tilde{i}}_{test}(s | \epsilon = 0.5)$. The policy $\rho^{\tilde{i}}_{test}$ is adversary $\tilde{i}$'s policy which is trained with $\pi_{test}$ in RMAAC, for all $\tilde{i} = \tilde{1}, \cdots, \tilde{N}$. We use $\rho_{test}$ to denote the joint policy of adversaries which is used in the testing. In summary, we test agents' joint policy $\pi^{scenario}_{test}(\tilde{s})$ when adversaries adopt the joint policy $\rho^{scenario}_{test}(s | \epsilon)$, and $\epsilon = 0.01, 0.05, 0.1, 0.5, 1, 2$, $scenario =$ Cooperative communication (CC), Cooperative navigation (CN), Predator Prey (PP), respectively. The testing is conducted over 400 episodes, and each episode has 25 time steps.

**Testing Results:** In Figures 18, 19 and 20, we compare the performance of RMAAC, M3DDPG, and MADDPG in scenarios CC, CN, and PP with different values of constraint parameters. MADDPG is a MARL baseline algorithm. M3DDPG is a robust MARL baseline algorithm. The y-axis denotes the mean episode reward of the agents.

From these figures, we can see that in all three scenarios, our RMAAC policies outperform the baseline MARL and robust MARL policies in terms of mean episode testing rewards under the attacks of linear noise format with different constraint parameters $\epsilon$. Our proposed RMAAC algorithm is robust to the state information attacks of linear noise format with different constraint parameters.

### C.3.3 Training Results Using Gaussian Noise with Different Variance

**Training Setup:** In this subsection, we train several RMAAC policies using Gaussian noise format as the state perturbation function, i.e. $f_2(s, b^{\tilde{i}}) = s + \mathcal{N}(b^{\tilde{i}}, \sigma)$. The variance $\sigma$ is respectively set as $0.001, 0.05, 0.1, 0.5, 1, 2$ and 3, given other hyper-parameters unchanged. Other used hyper-parameters can be found in Table 4.

**Training Results:** In Figure 21, 22 and 23, we show the training process of RMAAC in three scenarios: Cooperative communication (CC), Cooperative navigation (CN), Predator Prey (PP). The y-axis denotes the mean episode rewards of the agents and the x-axis denotes the training episodes.

From the figures, we can see that, in general, the smaller the value of variance used, the higher the mean episode rewards RMAAC can achieve. However, RMAAC has different sensitivities to the value of variance in different scenarios. When we use $\sigma = 3$, the RMAAC policies have the lowest mean episode rewards in all three scenarios. Nevertheless, when we use the smallest magnitude 0.001, the trained RMAAC policies do not always achieve the highest mean episode rewards. In these three scenarios, it is clear to see the performance of RMAAC using $\sigma = 1$ is better than or close to that of using $\sigma = 2$, and better than that of using $\sigma = 3$, i.e. Performance($\sigma = 1$) $\geq$ Performance($\sigma = 2$) $>$ Performance($\sigma = 3$). The performance of the RMAAC policies is close when the constraint parameters are less than or equal to 0.5.

### C.3.4 Testing Results Using Gaussian Noise with Different Variance

In this subsection, we test well-trained RMAAC policies in perturbed environments where adversaries adopt Gaussian noise format and different variances.

**Testing Setup:** The tested policy $\pi_{test}$ is trained with Gaussian noise format $f_2(s, b^{\tilde{i}}) = s + \mathcal{N}(b^{\tilde{i}}, \sigma = 1)$, constraint parameter is 0.5, where $b^{\tilde{i}} = \rho^{\tilde{i}}_{test}(s | \epsilon = 0.5)$. $\rho^{\tilde{i}}$ is adversary $i$'s policy which is trained with $\pi_{test}$ in RMAAC, for all $i = 1, \cdots, N$. We use $\rho_{test}$ to denote the joint policy of adversaries. In summary, we test agents' joint policy $\pi^{scenario}_{test}(\tilde{s})$ when adversaries adopt the joint policy $\rho^{scenario}_{test}(s | \epsilon = 0.5)$ and

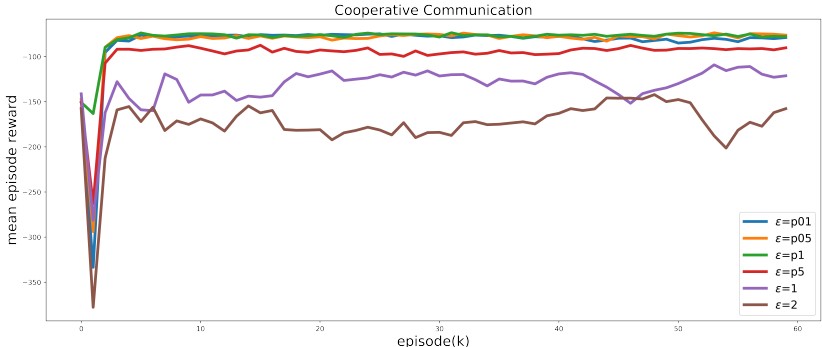

Figure 15: We train RMAAC policies using different values of constraint parameters in the scenario Cooperative Communication. In general, the smaller the constraint parameter is used, the higher the mean episode rewards RMAAC can achieve.

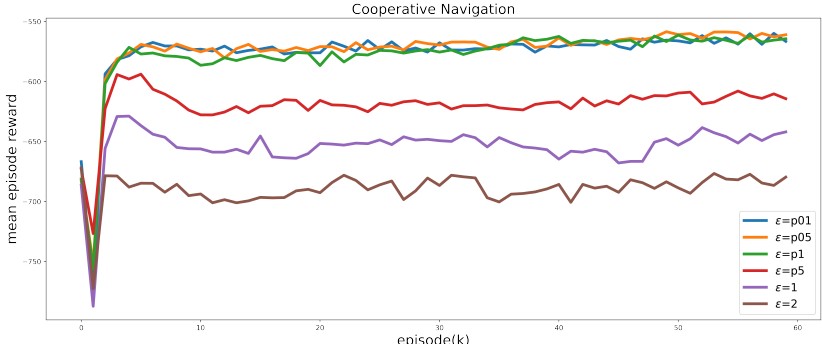

Figure 16: We train RMAAC policies using different values of constraint parameters in the scenario Cooperative Navigation. In general, the smaller the constraint parameter is used, the higher the mean episode rewards RMAAC can achieve.

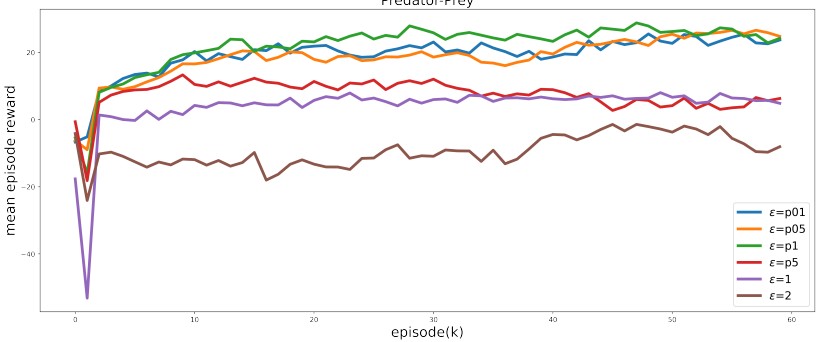

Figure 17: We train RMAAC policies using different values of constraint parameters in the scenario Predator-Prey. In general, the smaller the constraint parameter is used, the higher the mean episode rewards RMAAC can achieve.

Gaussian noise format $f_2(s, b^{\tilde{i}}) = s + \mathcal{N}(b^{\tilde{i}}, \sigma)$, where $\sigma = 0.001, 0.05, 0.1, 0.5, 1, 2, 3$, $scenario =$ Cooperative communication (CC), Cooperative navigation (CN), Predator Prey (PP). The testing is conducted over 400 episodes, and each episode has 25 time steps.

**Testing Results:** In Figures 24, 25 and 26, we respectively compare the performance of RMAAC, M3DDPG, and MADDPG in scenarios Cooperative communication, Cooperative navigation and Predator Prey with different values of constraint parameters. MADDPG is a MARL baseline algorithm. M3DDPG is a robust MARL baseline algorithm. The y-axis denotes the mean episode reward of the agents.

From these figures, we can see that in all three scenarios with all different values of constraint parameters, our RMAAC policies outperform the MARL and robust MARL baseline policies in terms of mean episode rewards under the attacks of Gaussian noise format with different variance. Our proposed RMAAC algorithm is robust to the state information attacks of Gaussian noise format with different values of variance.

### C.3.5 Testing Results under Different State Perturbation Functions

In this subsection, we test the well-trained RMAAC policies in perturbed environments where adversaries adopt different noise formats and policies.

**Testing Setup:** The tested agents' joint policy $\pi_{test}$ is trained with Gaussian noise format $f_2(s, b^{\tilde{i}}) = s + Gaussian(b^{\tilde{i}}, \sigma = 1)$, constraint parameter is 0.5, where $b^{\tilde{i}} = \rho_{test}^{\tilde{i}}(s|\epsilon = 0.5)$. $\rho_{test}^{\tilde{i}}$ is adversary $\tilde{i}$'s policy which is trained with $\pi_{test}$ in RMAAC, for all $\tilde{i} = \tilde{1}, \cdots, \tilde{N}$. We use $\rho_{test}$ to denote the joint policy of adversaries. In a summary, we test agents' joint policy $\pi_{test}^{scenario}(\tilde{s})$ when adversaries adopt the joint policy $\rho_{test}^{scenario}(s|\epsilon = 0.5)$, in three scenarios $scenario = $ Cooperative communication (CC), Cooperative navigation (CN), Predator Prey (PP), under non-optimal Gaussian format $f_3$, Uniform noise format $f_4$, fixed Gaussian noise format $f_5$ and Laplace noise format $f_6$, respectively. These noise formats are defined in the following:

$$
\begin{aligned}
f_3(s, b^{\tilde{i}}) &= s + Gaussian(b^{\tilde{i}}, 1) \quad \text{where} \quad b^{\tilde{i}} = \rho_{non-optimal}^{\tilde{i}}(s|\epsilon), \\
f_4(s, b^{\tilde{i}}) &= s + Uniform(-\epsilon, +\epsilon), \\
f_5(s, b^{\tilde{i}}) &= s + Gaussian(0, 1), \\
f_6(s, b^{\tilde{i}}) &= s + Laplace(b^{\tilde{i}}, 1) \quad \text{where} \quad b^{\tilde{i}} = \rho_{test}^{\tilde{i}}(s|\epsilon),
\end{aligned}
\tag{41}
$$

where $\rho_{non-optimal}^{\tilde{i}}$ is a non-optimal policy of adversary $\tilde{i}$. $\rho_{non-optimal}^{\tilde{i}}$ is randomly chosen from the training process. As we can see that $f_3$ and $f_6$ are independent of the optimal joint policy of adversaries, but $f_4$ and $f_6$ are not. The testing is conducted over 400 episodes, and each episode has 25 time steps.

**Testing Results:** In Figures 27, 28, and 29, we compare the performance of RMAAC, M3DDPG and MADDPG in scenarios Cooperative communication, Cooperative navigation and Predator Prey under 4 different noise formats, respectively. MADDPG is a MARL baseline algorithm. M3DDPG is a robust MARL baseline algorithm. The y-axis denotes the mean episode reward of the agents.

As we can see from these figures, most of the time, our RMAAC policy outperforms the MARL (MADDPG) and robust MARL (M3DDPG) baseline policies. In Cooperative communication and Predator Prey, under all 4 different noise formats, RMAAC policies achieve the highest mean episode rewards. In Cooperative navigation, RMAAC policies have the highest mean episode rewards when the non-optimal Gaussian noise format and Laplace noise format are used. The only exception happens in Cooperative navigation when the Uniform noise format and fixed Gaussian noise format are used. However, we can find that the performance of RMAAC policies is close to that of the baseline policies in terms of mean episode testing rewards. In general, our RMAAC algorithm is robust to different types of state information attacks.

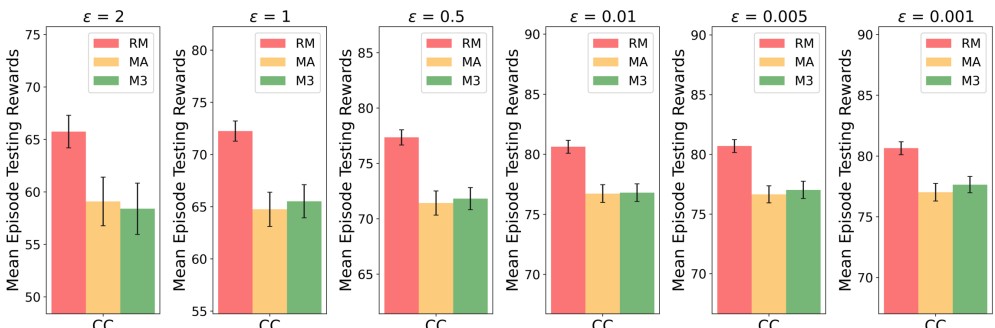

Figure 18: We test the performance of RMAAC(RM), MADDPG(MA), and M3DDPG(M3) policies under the attack of linear noise format when using different values of constraint parameters in the scenario Cooperative Communication. RM denotes our robust MARL algorithm, i.e. RMAAC. MA denotes MADDPG, a MARL baseline algorithm. M3 denotes M3DDPG, a robust MARL baseline algorithm. Our RMAAC algorithm outperforms baseline algorithms in terms of mean episode testing rewards under all situations using different values of constraint parameters.

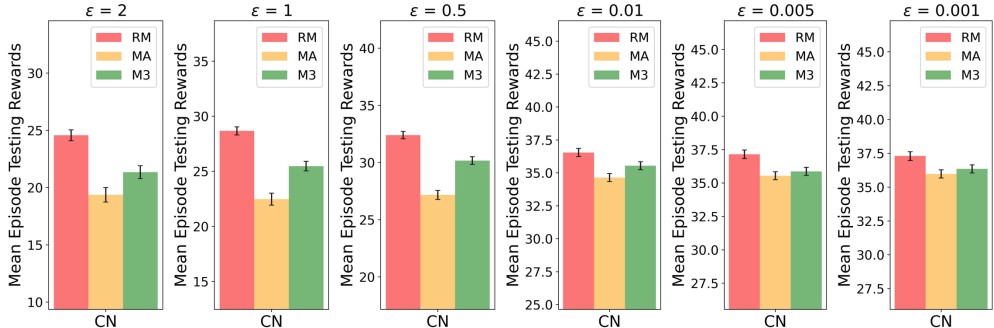

Figure 19: We test the performance of RMAAC(RM), MADDPG(MA), and M3DDPG(M3) policies under the attack of linear noise format when using different values of constraint parameters in the scenario Cooperative Navigation. RM denotes our robust MARL algorithm, i.e. RMAAC. MA denotes MADDPG, a MARL baseline algorithm. M3 denotes M3DDPG, a robust MARL baseline algorithm. Our RMAAC algorithm outperforms baseline algorithms in terms of mean episode testing rewards under all situations using different values of constraint parameters.

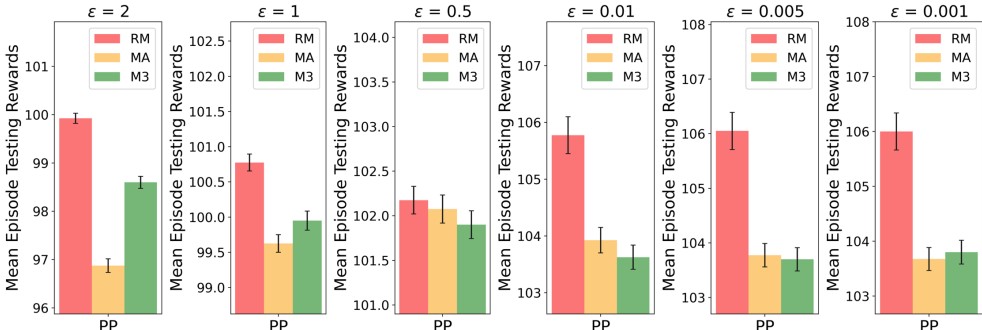

Figure 20: We test the performance of RMAAC(RM), MADDPG(MA), and M3DDPG(M3) policies under the attack of linear noise format when using different values of constraint parameters in the scenario Predator-Prey. RM denotes our robust MARL algorithm, i.e. RMAAC. MA denotes MADDPG, a MARL baseline algorithm. M3 denotes M3DDPG, a robust MARL baseline algorithm. Our RMAAC algorithm outperforms baseline algorithms in terms of mean episode testing rewards under all situations using different values of constraint parameters.

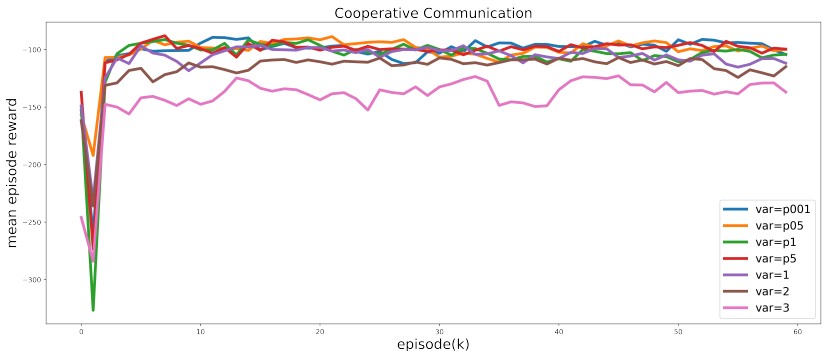

Figure 21: We train RMAAC policies using different values of variance in the scenario Cooperative Communication. In general, the smaller the variance is used, the higher the mean episode rewards RMAAC can achieve.

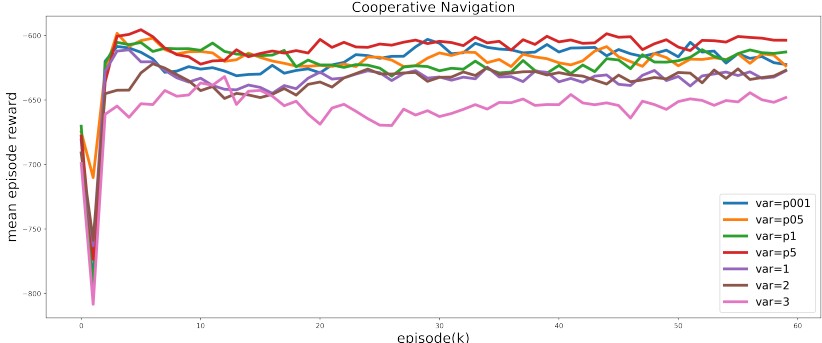

Figure 22: We train RMAAC policies using different values of variance in the scenario Cooperative Navigation. In general, the smaller the variance is used, the higher the mean episode rewards RMAAC can achieve.

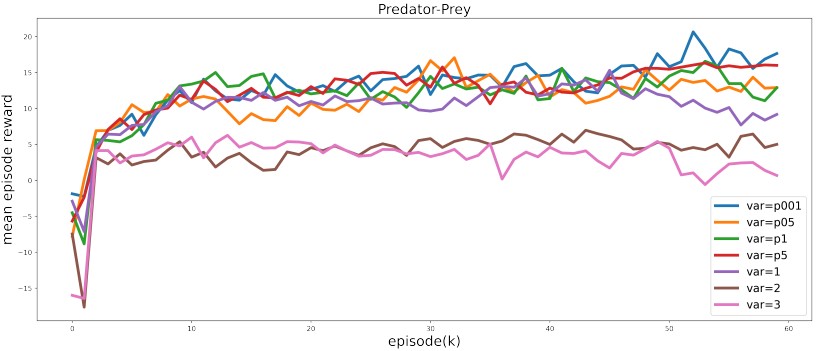

Figure 23: We train RMAAC policies using different values of variance in the scenario Predator-Prey. In general, the smaller the variance is used, the higher the mean episode rewards RMAAC can achieve.

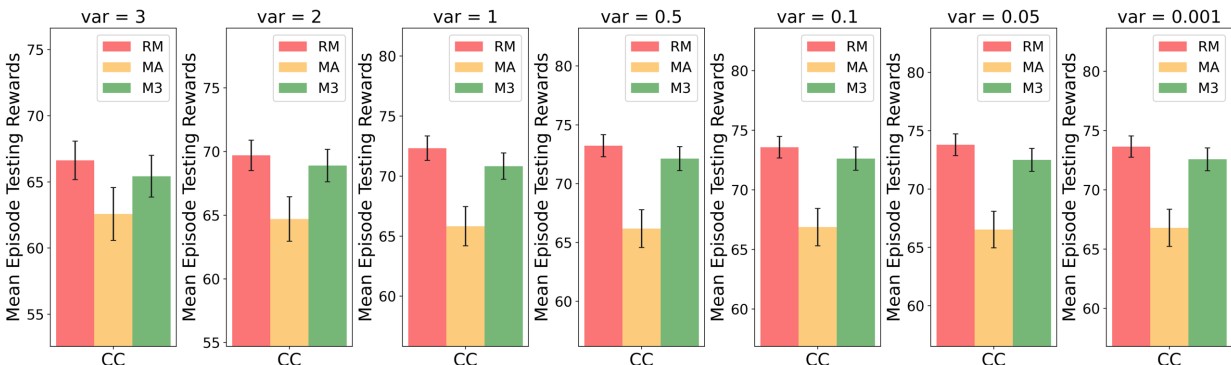

Figure 24: We test the performance of RMAAC(RM), MADDPG(MA), and M3DDPG(M3) policies under the attacks of Gaussian noise format with different variances in the scenario Cooperative Communication. RM denotes our robust MARL algorithm, i.e. RMAAC. MA denotes MADDPG, a MARL baseline algorithm. M3 denotes M3DDPG, a robust MARL baseline algorithm. Our RMAAC algorithm outperforms baseline algorithms in terms of mean episode testing rewards under all Gaussian noise formats with different variances.

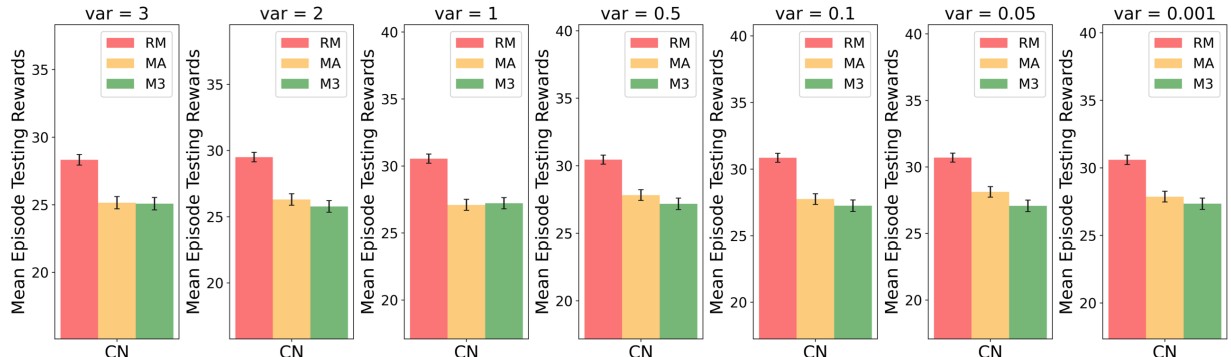

Figure 25: We test the performance of RMAAC(RM), MADDPG(MA), and M3DDPG(M3) policies under the attacks of Gaussian noise format with different variances in the scenario Cooperative Navigation. RM denotes our robust MARL algorithm, i.e. RMAAC. MA denotes MADDPG, a MARL baseline algorithm. M3 denotes M3DDPG, a robust MARL baseline algorithm. Our RMAAC algorithm outperforms baseline algorithms in terms of mean episode testing rewards under all Gaussian noise formats with different variances.

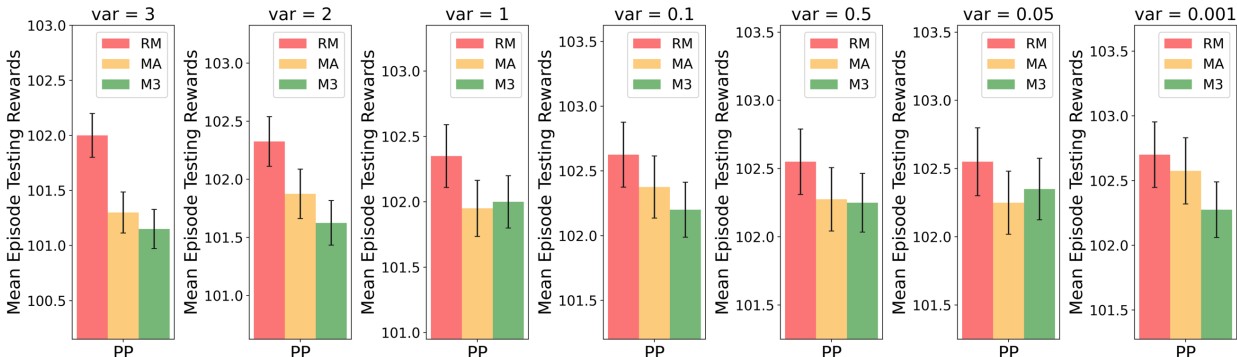

Figure 26: We test the performance of RMAAC(RM), MADDPG(MA), and M3DDPG(M3) policies under the attacks of Gaussian noise format with different variances in the scenario Predator-Prey. RM denotes our robust MARL algorithm, i.e. RMAAC. MA denotes MADDPG, a MARL baseline algorithm. M3 denotes M3DDPG, a robust MARL baseline algorithm. Our RMAAC algorithm outperforms baseline algorithms in terms of mean episode testing rewards under all Gaussian noise formats with different variances.

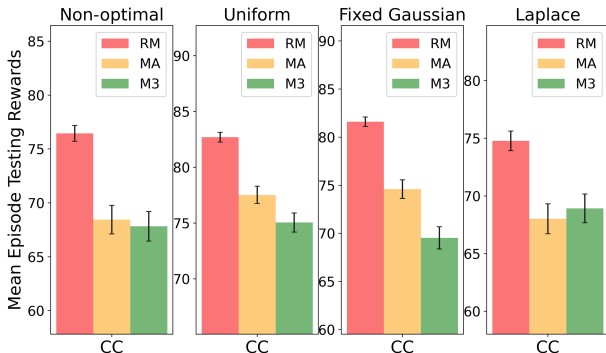

Figure 27: We test the performance of RMAAC(RM), MADDPG(MA), and M3DDPG(M3) policies under the attacks of different noise formats in the scenario Cooperative Communication. RM denotes our robust MARL algorithm, i.e. RMAAC. MA denotes MADDPG, a MARL baseline algorithm. M3 denotes M3DDPG, a robust MARL baseline algorithm. Our RMAAC algorithm outperforms baseline algorithms in terms of mean episode testing rewards under all kinds of attacks.

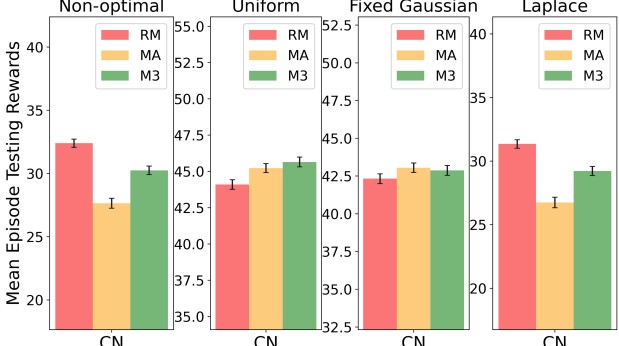

Figure 28: We test the performance of RMAAC policies under the attacks of different noise formats in the scenario Cooperative Navigation. RM denotes our robust MARL algorithm, i.e. RMAAC. MA denotes MADDPG, a MARL baseline algorithm. M3 denotes M3DDPG, a robust MARL baseline algorithm. Our RMAAC algorithm either outperforms or is close to baseline algorithms in terms of mean episode testing rewards under all kinds of attacks.

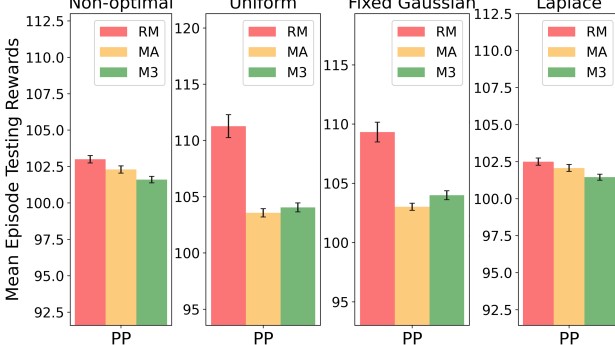

Figure 29: We test the performance of RMAAC policies under the attacks of different noise formats in the scenario Predator-Prey. RM denotes our robust MARL algorithm, i.e. RMAAC. MA denotes MADDPG, a MARL baseline algorithm. M3 denotes M3DDPG, a robust MARL baseline algorithm. Our RMAAC algorithm outperforms baseline algorithms in terms of mean episode testing rewards under all kinds of attacks.

