# OpenReview forum: "Robust Multi-Agent Reinforcement Learning with State Uncertainty"
_TMLR — Accepted by TMLR_

### Review · Reviewer_BAPE · 2023-03-27

**Summary Of Contributions:**

This paper investigates the problem of robust MARL under potential adversarial perturbations on the state space. To analyze this problem, a model MG-SPA is built and a robust equilibrium is proposed as the solution concept. Furthermore, the paper provides some structural results on the existence of the equilibrium and a convergent algorithm based on Q-learning. To scale to high-dimensional continuous problems, an empirical actor-critic style algorithm is also proposed. Empirical results also verify the effectiveness of the algorithm.

**Audience:**

Yes

**Claims And Evidence:**

Yes

**Requested Changes:**

1. Make the notation more clear. For example, in the beginning, the policy uses only the state as input. Later on, the policy receives the history as input. Please make the difference clear to readers.

2. Discussion on the reduced case, where there is only one agent is very beneficial.


**Strengths And Weaknesses:**

- Strengths

1. This is an interesting problem, where its single-agent counterpart has been studied extensively. It is important to see some difference arising up in the multi-agent settings.

2. The whole framework seems complete with well-motivated theoretical results, leading to practical algorithms.

- Weaknesses

1. For a robustness problem, it is not quite reasonable to consider a Nash equilibrium as a solution concept. Since what we usually care about is a robust victim, which is weaker than a Nash when minmax is not equal to maxmin. In my opinion, the condition for the maxmin to exist is much milder than the Nash to exist.

2. Following the first point, It seems only mixed robust NE is proved to exist. However, it seems the algorithm still tries to find a pure NE? In other words, in algorithmic design, we should hope to get a mixture of policy instead of a single policy? On the other hand, why not relax the solution concept to maxmin? If the current algorithm works for NE, we could hope it still works for maxmin.

3. When the system has only one agent and one adversary, what would the current analysis and algorithm reduce to? Does it lead to something we already know about single-agent robust RL or something still new to people? Some discussions or remarks will be quite helpful.

---

### Review · Reviewer_6cE4 · 2023-04-09

**Summary Of Contributions:**

The contributions of this paper are:

- Introducing the problem of multi-agent reinforcement learning (MARL) with state uncertainty caused by inaccurate sensing or state perturbation adversaries.

- Modeling the problem as a Markov Game with state perturbation adversaries (MG-SPA) and proposing robust equilibrium (RE) as the solution concept.

- Conducting a fundamental analysis of MG-SPA and giving conditions under which such an equilibrium exists.

- Proposing a robust multi-agent Q-learning (RMAQ) algorithm to find such an equilibrium, with convergence guarantees.

- Designing a robust multi-agent actor-critic (RMAAC) algorithm based on an analytical expression of the policy gradient derived in the paper to handle high-dimensional state-action space.

- Showing through experiments that the proposed RMAQ algorithm converges to the optimal value function and the RMAAC algorithm outperforms several MARL methods that do not consider state uncertainty in several multi-agent environments.

**Audience:**

Yes

**Broader Impact Concerns:**

I think there is no related ethical concerns about this paper.

**Claims And Evidence:**

Yes

**Requested Changes:**

Overall, I am satisfied with this paper. Maybe the authors address my concerns about the weakness of this paper.

**Strengths And Weaknesses:**

Pros:

- This paper addresses an important and challenging problem in multi-agent reinforcement learning, which is the robustness of agents' policies in the presence of state uncertainties.

- The authors provide a theoretical framework for analyzing the problem and proposes two algorithms, RMAQ and RMAAC, to find robust equilibria.

- The experiments show that the proposed algorithms outperform several existing MARL methods in several multi-agent environments. The paper also provides insights into solving an MG-SPA by solving a corresponding EFG, which may be useful for future research.

Cons:

- The paper does not consider the case where agents have different levels of sensing capabilities or different types of sensing errors.

- It also assumes that the state perturbation adversaries have complete knowledge of the agents' policies, which may not be realistic in some scenarios.

- Additionally, the paper focuses on discrete state and action spaces, and it may not be directly applicable to continuous state and action spaces.

- Finally, the experiments are limited to a few specific environments, and it would be interesting to see how the proposed algorithms perform in other scenarios.

---

### Review · Reviewer_VChK · 2023-04-12

**Summary Of Contributions:**

This work formalises and analyses the setting of learning in multi-agent systems under imperfect or adversely tampered state information. It introduces the framework of Markov Games with state perturbation adversaries (MG-SPA), together with robust equilibrium (RE) as a solution concept. It then also proposes two algorithmic contributions: robust multi-agent Q-learning (RMAQ), with convergence guarantees, as well as robust multi-agent actor-critic (RMAAC) for high-dimensional state-action spaces.

**Audience:**

Yes

**Broader Impact Concerns:**

Robustness in MARL is a valuable contribution for potentially sensible application domains, but perhaps a Broader Impact Statement is not required yet at this stage of development.

**Claims And Evidence:**

Yes

**Requested Changes:**

- Assumption 4.4 - more clear formatting (perhaps each assumption on a new line)
- A better connection/explanation on motivating the necessary assumptions for the theoretical analysis and results
- Additional discussions regarding the chosen formalism and policy types
- Additional context around the idea of history-based policies
- I also find it interesting to include in the main text the results and discussion on the performance of robust MARL approaches in worst-case settings and their potential degradation under lighter perturbations (pages 33, 34)
- Can you clarify how RMAAC is trained for the results in table 2 and figure 6, it is trained under the corresponding f?
- Could you provide a list of empirical assumptions required for training RMAAC, as well as parameters, eg: f, epsilon?
- If possible, would it be possible to add proof sketches/intuitions in the main text of the paper, to improve flow and readability?

Minor remarks:
- page 3 "where all players (...) use policies that no one has an incentive to deviate" $\rightarrow$ from which no one has...
- page 7 Section 4.3 "agent $i$ choose its actions" $\rightarrow$ chooses
- page 12 "In a summary" $\rightarrow$ In summary
- page 25 "We re-write Lemma it in Lemma...." $\rightarrow$ re-write it in ....


**Strengths And Weaknesses:**

Strengths:
- The work  nicely motivates and addresses the issue of state uncertainty in multi-agent reinforcement learning, focusing on the challenge of learning robust behaviours when faced with adversely tampered states.
-  The work provides a problem formulation, solution concept and algorithmic design for this setting, all in all a well-rounded contribution

Weaknesses:
- My main concern regards how appropriate the selected formalism is for this setting, namely the MG-SPA, where the perturbations are modelled using a functional form. While I understand that this choice allows one to stand on more 'solid' theoretical grounds, I believe that this setting warrants and can benefit from frameworks such as Dec-POMDPs or POSGs (especially after observing the results on including the history-based policies). In order to amend this issue, perhaps a short discussion, or remark should be included.
- Secondly, in order to improve the clarity and flow of this work, I think Section 4.3 can benefit from a better context or connection with the rest of the paper, as it currently appears disconnected from the rest.
- As far as I can tell, there is an inconsistent use of baselines (for RMAAC): M3DDPG, MADDPG, MAPPO (present only on the Appendix?). Can you motivate the current setup and why not all the baselines are present in the evaluation?
- I also missed the connection between some of the assumptions and the theoretical results, example Assumption 4.4 (5), or a direct discussion on their necessity.
- I would additionally suggest a discussion on the restriction or necessity of deterministic versus stochastic policies (I only find one mention, on page 9, on parametrizing the policies as deterministic ones).
- Finally, as far as I know, there is no page limit for the submission (as currently mentioned on page 12), so improving and expanding the evaluation will strengthen and improve the work.

---

### Public Comment · ~Ezgi_Korkmaz2 · 2023-08-01
**Problems with Robust Deep Reinforcement Learning**

It would be reasonable for this paper to refer to recent studies [1,2,3,4] on adversarial deep reinforcement learning. When certified adversarial training methods are explicitly referred to, it should also be mentioned that the recent studies have already shown that certified adversarial training techniques are vulnerable to many different sets of attacks from perturbations that can transfer [3], to natural directions [2]; furthermore, there have been recent solutions focusing on detecting adversarial directions to tackle robustness in deep reinforcement learning [1]. This study could acknowledge and refer to these studies.

[1]  Detecting Adversarial Directions in Deep Reinforcement Learning to Make Robust Decisions, ICML 2023.

[2] Adversarial Robust Deep Reinforcement Learning Requires Redefining Robustness. AAAI Conference on Artificial Intelligence, 2023.

[3] Deep Reinforcement Learning Policies Learn Shared Adversarial Features Across MDPs. AAAI Conference on Artificial Intelligence, 2022.

[4] Investigating Vulnerabilities of Deep Neural Policies. Conference on Uncertainty in Artificial Intelligence (UAI), Proceedings of Machine Learning Research (PMLR), 2021.

---

### Decision · Action_Editors · 2023-05-23

**Recommendation:** Accept as is

**Comment:**

This paper address the important problem of robustness in multiagent reinforcement learning (MARL). Uncertainty in the environment is modelled as "state perturbation adversaries": agents that can see the full state and can take actions that corrupt the observation of the state for the other players.

All the reviewers agree toward accepting this paper for TMLR with no outstanding criticisms, other than a minor comment that authors should take all the reviewer feedback into account when updating the paper.

The authors were quite responsive in their discussions with reviewers and have updated the paper as a result of the feedback received. However, the authors should double-check that every comment is addressed before submitting the camera-ready final copy.


**Audience:**

This will interest the MARL community, which is well-represented in TMLR's audience.

**Claims And Evidence:**

The claims are well-supported by accurate, convincing, and clear evidence.

The proof expands the augmented Markov game to an extensive-form adversarial game which can be solved for the Nash equilibirum and mapped back, which is possible due to finite-horizon h and Assumption 4.4.

The experiments confirm that a robust equilibirum can be found by RMAQ in practice on an example game, and that the actor-critic algorithm   RMAAC is competitive on the larger multiplayer particle environments.